# Adaptation and selection shape clonal evolution of tumors during residual disease and recurrence

Andrea Walens [1,6], Jiaxing Lin[2,6], Jeffrey S. Damrauer [1,6], Brock McKinney[1], Ryan Lupo[1], Rachel Newcomb[1], Douglas B. Fox [1], Nathaniel W. Mabe [1], Jeremy Gresham[3], Zhecheng Sheng[3], Alexander B. Sibley [3], Tristan De Buysscher[4,5], Hemant Kelkar[5], Piotr A. Mieczkowski[4,5], Kouros Owzar[2,3] & James V. Alvarez [1✉]

The survival and recurrence of residual tumor cells following therapy constitutes one of the biggest obstacles to obtaining cures in breast cancer, but it remains unclear how the clonal composition of tumors changes during relapse. We use cellular barcoding to monitor clonal dynamics during tumor recurrence in vivo. We find that clonal diversity decreases during tumor regression, residual disease, and recurrence. The recurrence of dormant residual cells follows several distinct routes. Approximately half of the recurrent tumors exhibit clonal dominance with a small number of subclones comprising the vast majority of the tumor; these clonal recurrences are frequently dependent upon *Met* gene amplification. A second group of recurrent tumors comprises thousands of subclones, has a clonal architecture similar to primary tumors, and is dependent upon the Jak/Stat pathway. Thus the regrowth of dormant tumors proceeds via multiple routes, producing recurrent tumors with distinct clonal composition, genetic alterations, and drug sensitivities.

[1] Department of Pharmacology and Cancer Biology, Duke University, Durham, NC 27710, USA. [2] Department of Biostatistics and Bioinformatics, Duke University, Durham, NC 27710, USA. [3] Duke Cancer Institute, Duke University, Durham, NC 27710, USA. [4] Lineberger Comprehensive Cancer Center, University of North Carolina, Chapel Hill, NC 27599, USA. [5] Department of Genetics, University of North Carolina, Chapel Hill, NC 27599, USA. [6] These authors contributed equally: Andrea Walens, Jiaxing Lin, Jeffrey S. Damrauer. ✉email: james.alvarez@duke.edu

Tumor recurrence after initial treatment is a frequent cause of death in many cancers, including breast cancer, and recurrent tumors are often resistant to therapies to which the corresponding primary tumors were sensitive[1–4]. Recurrent tumors are thought to arise from a population of cells that survives initial treatment; these cells, often referred to as minimal residual disease, can persist in a clinically undetectable state for years or even decades before resuming growth to give rise to recurrent tumors[5–7]. In spite of the clinical importance of residual disease and recurrence, little is known about the pathways that regulate the long-term survival of residual cells or that induce the reactivation of these cells to yield recurrent tumors[8]. Identifying such pathways may suggest opportunities for directly killing residual cells or preventing their reactivation, thereby forestalling the development of recurrent tumors[9].

Primary breast tumors are heterogeneous, harboring different subclones of genetically or epigenetically distinct cells[10]. While tumor progression is thought to be driven by the progressive outgrowth of aggressive subclones, little is known about how the clonal composition of tumors changes during residual disease and recurrence. Understanding the clonal dynamics of dormancy and recurrence is essential for developing strategies to prevent or treat recurrence. However, studying these processes in patients is challenging, given the difficulty in identifying residual disease and obtaining recurrent tumors.

To overcome these obstacles, we and others have used an inducible genetically engineered mouse model that exhibits key features of breast cancer progression, including the survival of cancer cells following therapy and their eventual spontaneous recurrence[11–16]. In this model, doxycycline (dox) administration to MMTV-rtTA;TetO-neu (MTB;TAN) mice induces expression of the Her2/neu oncogene, leading to the formation of invasive mammary tumors. Subsequent withdrawal of dox induces Her2 downregulation and complete tumor regression, perhaps mimicking anti-Her2 targeted therapy. However, a small population of residual cells survives Her2 downregulation and persists in a dormant, non-proliferative state in the mammary gland[16]. These residual cells eventually reinitiate proliferation, independent of Her2 expression, to form a recurrent tumor. Here we combine this conditional mouse model with lentiviral-mediated cellular barcoding to study the clonal dynamics of tumor regression, residual disease, and relapse. We find a progressive decrease in clonal complexity during tumor regression, residual disease, and recurrence. Only a fraction of subclones survives oncogene withdrawal and persists in residual tumors. The minimal residual disease phase itself is accompanied by a continued attrition of clones, suggesting an ongoing process of selection during dormancy. The reactivation of dormant residual cells into recurrent tumors follows several distinct evolutionary routes. Approximately half of the recurrent tumors exhibit a striking clonal dominance in which one or two subclones comprise the vast majority of the tumor. The majority of these clonal recurrent tumors exhibit evidence of de novo acquisition of Met amplification and are sensitive to small-molecule Met inhibitors. A second group of recurrent tumors exhibits marked polyclonality, with thousands of subclones and a clonal architecture very similar to primary tumors. These polyclonal recurrent tumors are not sensitive to Met inhibitors, but are instead dependent upon an autocrine IL-6—Jak/Stat3 pathway. These results identify diverse mechanism of tumor recurrence and suggest that understanding the clonal dynamics of relapse may help inform strategies to prevent or treat recurrent tumors.

## Results

### Cellular barcoding to track clonal dynamics during tumor regression, residual disease, and recurrence. We used a cellular

barcoding strategy to directly monitor changes in the clonal composition of tumors during tumor regression, residual disease, and recurrence. In this approach, random, inert DNA barcodes are introduced into a population of cells. Each barcode serves as a molecular tag that marks all of the progeny of a particular cell. Consistent with the nomenclature used in other studies[17–29], all cells marked by a particular barcode are referred to here as a clone, reflecting the fact that they arose from a single cell. It is important to note that the term clone is not meant to imply genetic differences between cells marked by different barcodes. The clonal composition of a population can be determined by measuring the number of clones (number of unique barcodes detected) and the abundance of each clone (proportion of total reads contributed by each barcode) using next-generation sequencing. In this manner, it is possible to track the clonal dynamics of a cell population under different conditions. This approach has been used to study clonal dynamics in response to targeted therapies in vitro[17–19] and during the growth and metastatic spread of tumors in vivo[20–29].

To implement this barcoding strategy we digested a primary Her2-driven tumor (donor tumor #1) from an MTB;TAN mouse and cultured tumor cells ex vivo in the presence of dox to maintain Her2 expression. Approximately 200,000 tumor cells were then infected with a lentiviral barcode library at a multiplicity of infection of 0.1 to ensure that each cell received a single DNA barcode. Following selection, the population was expanded for 12 population doublings, yielding a population of 100 million cells containing ~20,000 unique barcodes with each barcode represented in an average of 5000 cells (Fig. 1a). A recent report highlighted the importance of ensuring that barcode composition is stable in vitro prior to injecting barcoded cells into mice[28]. Therefore, we passaged these cells for an additional ten population doublings and sequenced the barcodes in the final population. We observed only a modest decrease in both the number of unique barcodes and the Shannon Diversity Index following ten population doublings, and the barcode composition was very similar to the starting population, suggesting that the injected cell population was stable in vitro (Supplementary Fig. 1A–D).

The barcoded cell population was injected (1 million cells per injection, 50 cells per barcode) into the mammary glands of a large cohort of recipient mice on dox (Fig. 1b). Primary tumors formed ~3 weeks following injection (Fig. 1c). Once primary tumors reached 5 mm in diameter, one cohort of mice was sacrificed with primary tumors ($n = 6$ tumors), and the remaining mice were removed from dox to induce Her2 downregulation, leading to rapid tumor regression (Fig. 1c). Additional cohorts of mice were sacrificed with residual tumors at 4 and 8 weeks following dox withdrawal ($n = 6$ tumors per cohort), prior to the time point at which most mice develop recurrent tumors (Fig. 1c, d). (Note that while some mice develop recurrent tumors by 8 weeks post-dox withdrawal, these tumors were excluded from the residual tumor cohort; all residual tumors were confirmed to be smaller than 3 mm in diameter upon sacrifice.) A final cohort of mice was allowed to develop recurrent tumors ($n = 12$ tumors); recurrent tumors in these mice arose with a median time of 75 days (Fig. 1d). Importantly, this timing is similar to mice injected with primary tumor cells lacking DNA barcodes, suggesting that introduction of this barcode library does not affect the timing of recurrent tumor formation.

Primary and recurrent tumors arising in this orthotopic setting have a histology that resembles tumors from the autochthonous model. Primary tumors exhibited an epithelial morphology, with nests of epithelial cells surrounded by stromal cells (Fig. 1e, left). Recurrent tumors had a mesenchymal morphology, consistent with the previous finding that tumor recurrence in these models

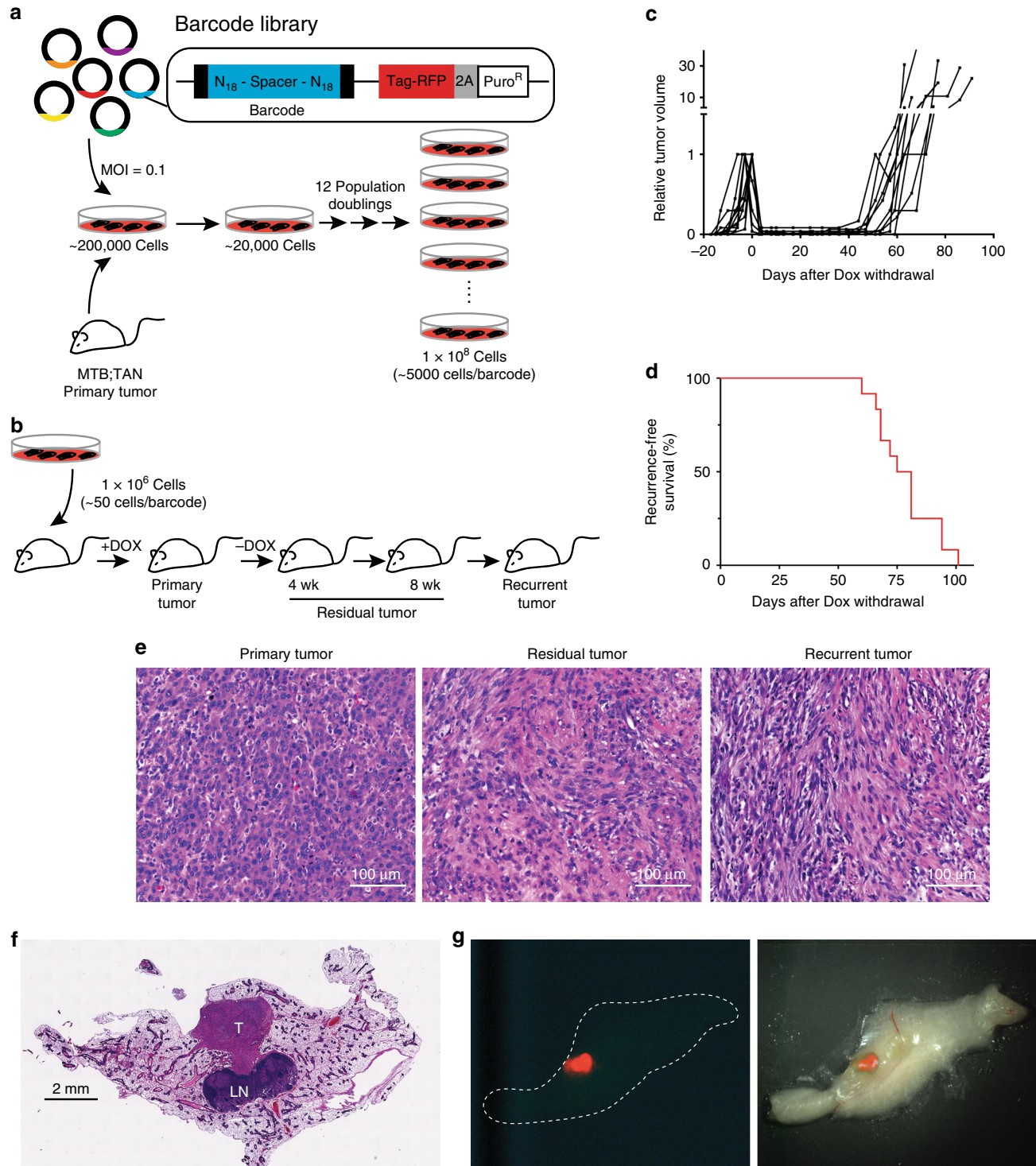

**Fig. 1 Cellular barcoding to track clonal dynamics during tumor regression, residual disease, and recurrence. a** Schematic of cellular barcoding strategy. Primary Her2-driven mouse mammary tumors were digested, cultured with Dox, and 200,000 cells were infected at low MOI (0.1) with a lentiviral barcode library comprising ~60 million unique barcodes. Following selection for transduced cells, the population was expanded to yield a population of 100 million cells comprising ~20,000 unique barcodes (~5000 cells/barcode). **b** One million barcoded tumor cells were injected bilaterally into the inguinal mammary glands of nu/nu mice on Dox. Once orthotopic primary tumors reached 5 mm in diameter, one cohort of mice was sacrificed with primary tumors (*n* = 6 tumors). Dox was removed from the remaining mice, and cohorts were sacrificed with residual tumors after 4 weeks (*n* = 6 tumors) and 8 weeks (*n* = 6 tumors). A final cohort of mice was monitored until recurrent tumors formed (*n* = 12 tumors). **c** Tumor growth curves showing primary tumor growth, regression, residual disease, and recurrence for representative orthotopic Her2-driven tumors. **d** Kaplan–Meier recurrence-free survival curves for barcoded orthotopic tumors. **e** H&E-stained sections of a representative primary, residual, and recurrent tumor. Scale bar = 100 μm. Images are representative of four independent tumors. **f** H&E-stained section of a mammary gland whole-mount showing the size and location of a representative residual tumor. T tumor, LN lymph node. Scale bar = 2 mm. Image is representative of three independent tumors. **g** Fluorescent (left) and merged fluorescent-brightfield (right) image of a representative fluorescently labeled residual tumor (red) within the mammary gland.

is associated with epithelial-to-mesenchymal transition (EMT)[14] (Fig. 1e, right). Interestingly, residual tumors comprised cells with both epithelial and mesenchymal characteristics, and also had a notable stromal component (Fig. 1e, middle).

Because many recurrent breast cancers arise at distant sites, we also examined the behavior of lung metastases in response to Her2 downregulation. Mice on dox were injected in the tail vein with primary Her2-driven tumor cells (donor tumor #1) and luciferase imaging was used to measure the growth of lung metastasis. Mice developed lung metastases within 3 weeks (Supplementary Fig. 2A, B). Four weeks following tumor cell injection, one cohort of mice ($n = 4$) was left on dox, and the remaining mice ($n = 4$) had dox withdrawn from their drinking water to induce Her2 downregulation. Dox withdrawal led to Her2 downregulation as measured by luciferase imaging (Supplementary Fig. 2B). Mice left on dox had to be sacrificed due to moribundity at 5 weeks following injection, and these mice has abundant metastasis grossly visible in their lungs (Supplementary Fig. 2A, C). In contrast, mice removed from dox survived an additional 4 weeks and were then sacrificed. While we could not directly confirm that Her2 downregulation led to tumor regression, the observation that mice removed from dox survived longer than mice left on dox suggests that Her2 downregulation led to regression of metastases, as previously described[15]. Upon sacrifice, mice removed from dox had extensive lung metastases in the absence of Her2 expression (Supplementary Fig. 2B, D). Together, these data suggest that lung metastasis arising from Her2-driven tumors regress following Her2 downregulation, but rapidly recur through Her2-independent mechanisms.

**Primary tumors are driven by the expansion of a subset of clones**. To measure the barcode composition of tumors, we next isolated genomic DNA from tumors at each time point. To ensure accurate representation of barcodes, we isolated genomic DNA from the entire tissue sample for both primary and recurrent tumors. Although residual tumors were readily apparent on an H&E-stained section (Fig. 1f), they were too small to identify grossly. Therefore, to facilitate isolation of barcodes from these residual tumors, tumors were micro-dissected using a fluorescence microscope prior to DNA isolation (Fig. 1g). In this manner, we were able to isolate barcoded genomic DNA from primary, residual, and recurrent tumors for subsequent sequencing.

We used next-generation sequencing to measure the number of barcodes and their relative abundance in each tumor. For every sample, barcodes from ~100,000 cells were amplified and sequenced. The starting cell population contained ~17,000 unique barcodes (Fig. 2a, b), consistent with our target of 20,000 barcodes. The mean barcode frequency was 0.00588%, and the frequency of the most abundant barcode was 0.707% (range: 0.0000736–0.707%; Fig. 2a). This suggests that the growth of these cells in vitro did not impose a strong selection for individual clones. In contrast, an average of 4544 barcodes were detected in primary tumors (range: 3063–5648), and there was evidence for a selection for individual barcodes (Fig. 2a, b). To quantitatively assess the clonal complexity of each tumor we used the Shannon Diversity index[30], a measure of the diversity of a population that incorporates both the number of clones and their representation. Tumors with many clones that are evenly distributed have a high Shannon Index, while tumors with few or dominant clones have a low Shannon Index. This analysis revealed that primary tumors had a reduction in barcode complexity as compared to the starting population (Fig. 2c). Consistent with this, barcodes were more unevenly distributed in primary tumors: 50% of reads were contributed by an average of 56 barcodes in primary tumors, as

compared to 700 barcodes in the starting population (Fig. 2d). The most abundant barcode in primary tumors represented an average of 8.4% of the population (range: 4.5–12.8%), as compared to 0.707% in the starting population (Fig. 2e–g and Supplementary Fig. 3). Taken together, these results suggest that primary tumor formation is accompanied by a reduction in clonal complexity and the expansion of a subset of clones.

**Independent primary tumors have similar clonal composition**. The observation that the formation of primary tumors is associated with a reduction in clonal complexity suggests that tumor growth in vivo imposes different or more stringent selective pressures than growth in vitro. We considered two possibilities: abundant clones in primary tumors may have also been abundant in the starting population and simply expanded further during tumor growth in vivo. Alternatively, tumor growth may have selected for a distinct subset of clones that were not abundant in the starting population. To distinguish among these possibilities, we compared abundant clones between tumors and the starting population. The barcodes that were most abundant in primary tumors were not abundant in the injected cell population (Fig. 2g), suggesting that tumor growth in vivo selects for distinct clones. We next explored whether the same clones were most abundant in independent primary tumors, which would suggest that there were preexisting clones in the starting population with an inherent capacity to grow in vivo. Examination of the barcodes composing the top 15% of reads revealed that the same set of barcodes were most abundant in independent primary tumors, but not in the staring population of cells (Fig. 2h). For instance, barcode 8240:8518 was the most abundant in all primary tumors, and represented an average of 8.4% of total reads in primary tumors (range: 4.5–12.8%). In contrast, this barcode composed only 0.027% of reads in the starting cell population, indicating that the clone marked by this barcode expanded 300-fold during primary tumor growth. To quantitatively assess the similarity in clonal composition of tumors we calculated the Jensen–Shannon divergence[31] between barcode abundances in independent primary tumors. All primary tumors were highly similar to one another but dissimilar from the starting population of injected cells (Fig. 2i; $P$ value = $8.235 \times 10^{-7}$, Welsh Two-Sample $t$-test). Taken together, these results indicate that primary tumor growth selects for a subset of preexisting clones with increased tumorigenic potential, and highlights the utility of cellular barcoding to identify such clones.

These results suggest the presence of intratumoral heterogeneity in primary Her2-driven tumors. To directly assess this, we performed single-cell RNA-seq (scRNA-seq) on tumor cells derived from two independent primary tumors from MTB;TAN mice (donor #1 and donor #2). We sequenced cells cultured from tumors to avoid confounding effects of nontumor cells from the microenvironment. We also sequenced two recurrent tumor cell lines (recurrent tumor cell line #1 and #3) as comparators. We found that both primary tumor cell cultures had extensive transcriptional heterogeneity (Supplementary Fig. 4A). Primary tumor cell cultures had a small population of cells that were present in clusters enriched for recurrent tumor cells (Supplementary Fig. 4B–E). For instance, pairwise comparison of primary tumor cell line #1 with recurrent tumor cell line #3 identified a cluster (Cluster #7) distinguished by high expression of mesenchymal genes, including Vimentin, S100A6, and Timp1 (Supplementary Fig. 4C and Supplementary Data 1). While the majority of cells in this cluster were recurrent tumor cells, a small fraction of primary tumor cells was present in this cluster (Supplementary Fig. 4C). Similar results were found by performing pairwise comparison of

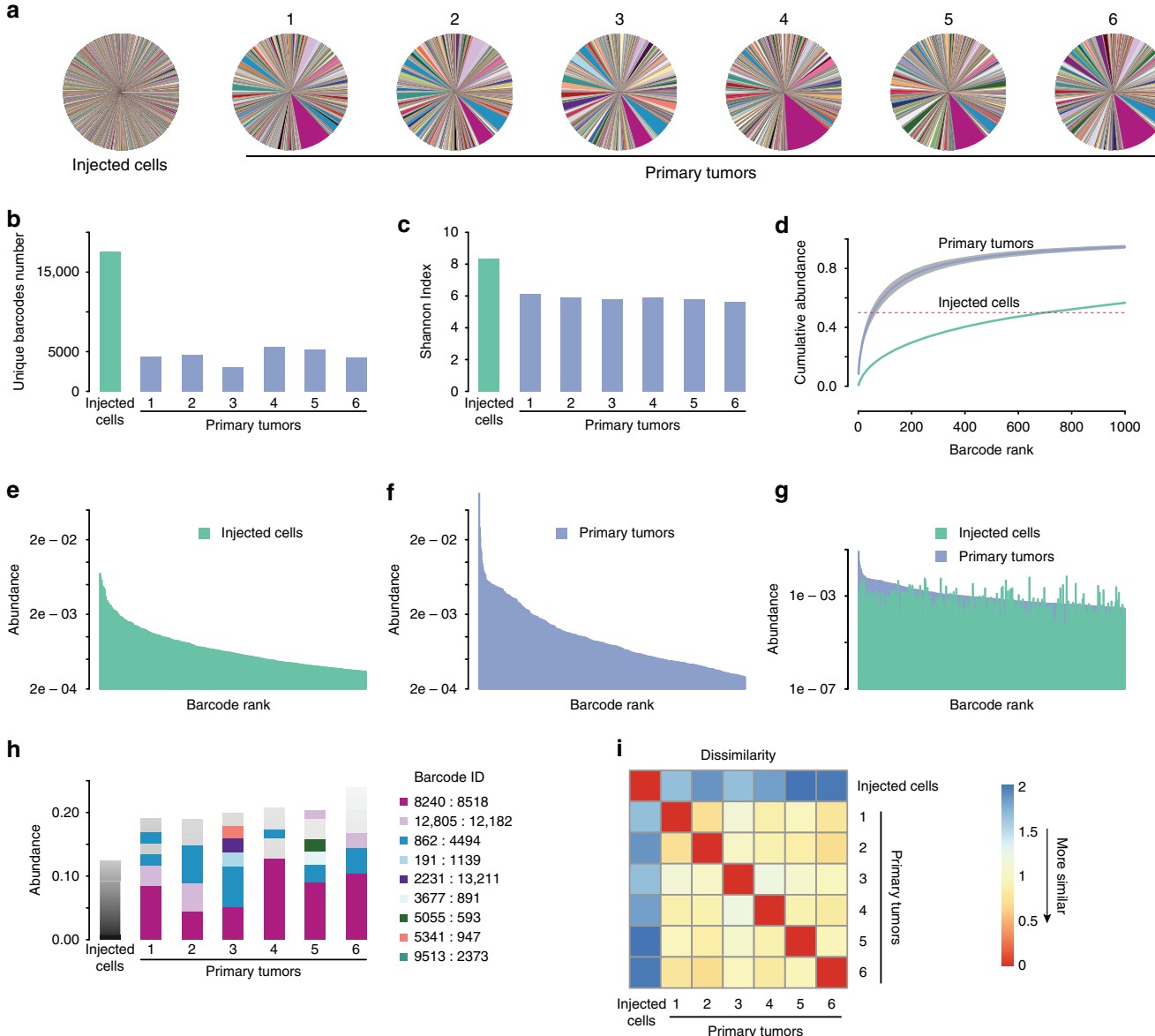

**Fig. 2 Independent primary tumors have similar clonal composition. a** Pie charts showing the relative frequency of barcodes in the starting cell population and six independent primary tumors. Note that individual barcodes are represented by the same color in each pie chart. **b** Number of unique barcodes detected in the starting cell population and six independent primary tumors. $n = 1$ for the injected cell population and $n = 6$ biologically independent primary tumors. **c** Shannon diversity index showing barcode complexity of the starting cell population and six independent primary tumors. $n = 1$ for the injected cell population and $n = 6$ biologically independent primary tumors. **d** Cumulative abundance plots for the starting cell population and the average of six primary tumors, demonstrating that barcode complexity decreases during primary tumor growth. **e** Barcode abundance in the starting cell population. Barcodes are ranked on the *x*-axis from most to least abundant. **f** Barcode abundance in a representative primary tumor. Barcodes are ranked on the *x*-axis from most to least abundant. **g** Comparison of barcode distribution in the starting cell population and a representative tumor, showing that the most abundant barcodes in tumors were not abundant in the starting population. Barcodes are ranked on the *x*-axis based on their abundance in the primary tumor. **h** A subset of barcodes is reproducibly enriched in independent primary tumors. The most abundant barcodes in each sample are shown, with each color denoting a unique barcode. Note that barcode colors match pie charts in **a**. $n = 1$ for the injected cell population and $n = 6$ biologically independent primary tumors. **i** Correlation matrix showing the similarity in barcode abundance between samples. Primary tumors had similar barcode distributions to one another, but were distinct from the starting population. The Jensen–Shannon divergence was used to measure dissimilarity among tumors.

primary tumor cell line #1 with recurrent tumor cell line #1 (Supplementary Fig. 4D, E). Clusters #2 and 3, which had high expression of mesenchymal genes including Vimentin, Timp1, and Twist1, were predominantly composed of recurrent tumor cells but had small populations of primary tumor cells (Supplementary Fig. 4D, E and Supplementary Data 2). Taken together, these results suggest that primary Her2-driven tumors have preexisting transcriptional heterogeneity which may serve

as a basis for the different behavior of subclones in these tumors.

**Reduction in clonal complexity during tumor regression and residual disease.** We next asked how the clonal composition of tumors changed following tumor regression and during residual disease. As described above, Her2 downregulation in primary

tumors leads to near complete tumor regression, but a small population of tumor cells survives and persists in a non-proliferative state (see Fig. 1f, g). Tumors were harvested at 4 weeks (early residual tumors) and 8 weeks (late residual tumors) following dox withdrawal. Comparing primary tumors to early residual tumors provides insight into changes in clonal composition during tumor regression, while comparisons between early and late residual tumors can reveal changes that occur during residual disease.

Sequencing of barcodes in tumors 4 weeks after dox withdrawal revealed that tumor regression was accompanied by a decrease in the clonal complexity of tumors and the further enrichment of a subset of clones (Fig. 3a). Early residual tumors had fewer barcodes (Fig. 3c; $P$ value $= 2.6 \times 10^{-3}$, Welsh Two-Sample $t$-test) and a reduction in barcode complexity (Fig. 3d; $P$ value $= 4.8 \times 10^{-4}$, Welsh Two-Sample $t$-test) as compared to primary tumors. Barcodes in these tumors were more unevenly distributed than primary tumors, with 50% of total reads coming from only ~16 barcodes (Fig. 3e; $P$ value $= 3.3 \times 10^{-4}$, Welsh Two-Sample $t$-test). Barcode 8240:8518—which was the most abundant in primary tumors—was further enriched following tumor regression, and composed on average of 17.4% of the population (range: 4.2–34.1%; Fig. 3a and Supplementary Fig. 5A). Taken together, these results indicate that tumor regression following Her2 downregulation is accompanied by a reduction in clonal complexity.

Following tumor regression, residual tumors persist for between 1 and 2 months before developing into recurrent tumors. We next determined how the clonal composition of tumors changes during this residual disease period by comparing barcode number and abundance between early and late residual tumors. Surprisingly, we found that there was a continued reduction in clonal complexity during this residual disease stage (Fig. 3a, b). Late residual tumors had fewer barcodes ($P$ value $= 2.4 \times 10^{-2}$, Welsh Two-Sample $t$-test), reduced Shannon index ($P$ value $= 2.0 \times 10^{-2}$, Welsh Two-Sample $t$-test), and a more uneven distribution of barcodes ($P$ value $= 8.0 \times 10^{-3}$, Welsh Two-Sample $t$-test) as compared to early residual tumors, with 50% of total reads coming from 5.5 barcodes on average (Fig. 3c–e). Barcode 8240:8518 remained the most abundant barcode in all late residual tumors, and represented as high as 49% of total reads (range: 10.4–49.2%; Fig. 3b and Supplementary Fig. 5A). Notably, the decrease in clonal complexity between 4 and 8 weeks post-dox withdrawal was similar in magnitude to the decrease in complexity that accompanied tumor regression (Fig. 3d).

We next considered whether the apparent reduction in complexity in residual tumors might be due to the failure to detect rare barcodes in these tumors. To address this, we performed a simulation experiment where we randomly sampled decreasing numbers of reads from each tumor (range: $2000$–$2 \times 10^8$) and calculated the Shannon Index for this reduced read number (Supplementary Fig. 5B–D). This analysis showed that the Shannon index for all tumors was largely unchanged over this wide range of reads (Supplementary Fig. 5B–D). All early and late residual tumors had a lower Shannon index than primary tumors irrespective of read number (Supplementary Fig. 5B–D), indicating that the reduced complexity in these tumors was not due to a failure to detect rare subclones. Taken together, these results suggest that the residual disease state is accompanied by a continued attrition of a subset of clones, perhaps indicative of ongoing selective pressures during residual disease.

To gain insight into how the overall clonal architecture of tumors changed during tumor regression and residual disease, we examined the similarity of barcode abundances between residual tumors and primary tumors based on the Jensen–Shannon

divergence. As shown above, primary tumors were highly similar to one another (Fig. 2i and Supplementary Fig. 5e). In contrast, the barcode composition of early and late residual tumors was progressively less similar to that of primary tumors (Supplementary Fig. 5e; primary tumors vs. early residual tumors, $P$ value $= 2.712 \times 10^{-15}$; primary tumors vs. late residual tumors, $P$ value $< 2.2 \times 10^{-16}$; Welsh Two-Sample $t$-test). Taken together, these results suggest that the clonal composition of tumors progressively changes during tumor regression and residual disease, with residual tumors exhibiting a greater divergence in their clonal composition.

Finally, we asked whether the changes in clonal composition during tumor regression were distinct from changes that would be observed as tumors continue to grow in the presence of Her2. To address this, we generated an additional cohort of mice in which tumors were allowed to grow in the presence of dox until they reached maximum tumor volume (~15 mm in diameter). On average, these late primary tumors were ninefold larger in volume than tumors in the original primary tumor cohort (data not shown). Late primary tumors had fewer unique barcodes and a lower Shannon Index compared to primary tumors (Supplementary Fig. 6A–C), suggesting that continued tumor growth is associated with decreased clonal complexity. However, the clonal composition of late primary tumors was more similar to primary tumors than to residual tumors as measured by the Jensen–Shannon divergence (Supplementary Fig. 6D). Based upon this, we conclude that both continued tumor growth in the presence of Her2, as well as tumor regression induced by Her2 downregulation, are accompanied by decreased clonal complexity. However, these different scenarios impose different selective pressures and therefore select for different clones.

**Distinct clonal architecture in recurrent tumors suggests divergent routes to recurrence.** The findings above indicate that tumor regression and residual disease were accompanied by a decrease in the number and complexity of clones. Nonetheless, all residual tumors still retained several thousand clones that could serve as a template for further selection. We reasoned that tumor recurrence could be driven by the reactivation of a small subset of clones, or by a broader, tumor-wide reactivation of many or most of the clones in residual tumors. To distinguish between these possibilities, we examined the clonal composition of recurrent tumors. We found that recurrent tumors exhibited a surprisingly large variation in the number and distribution of barcodes (Fig. 3f). Three recurrent tumors (recurrent tumors #1-3) contained thousands of barcodes that were relatively evenly distributed, as evidenced by a high-diversity index (Fig. 3f, g). At the other end of the spectrum, in three recurrent tumors (recurrent tumors #10-12) almost all reads were contributed from a single barcode (Fig. 3f). The grossly uneven distribution of barcodes in these clonal tumors was reflected by a very low Shannon Diversity Index (Fig. 3g). Interestingly, in recurrent tumor #10 and #12, the most abundant barcode, 8240:8518, was also the most abundant barcode in primary and residual tumors (Fig. 3f and Supplementary Fig. 5A). The remaining tumors (recurrent tumors #4-9) had an intermediate clonal complexity (Fig. 3f and g). Tumors #4, 6, 8, and 9 had between two and eight dominant barcodes, while tumors #5 and 7 had a single abundant barcode together with a large number of minor barcodes.

We next used the Jensen–Shannon divergence to assess how the clonal architecture of these recurrent tumors compared to primary tumors. The barcode distribution of tumors #1-3 was very similar to primary tumors, suggesting that the clonal composition of these recurrent tumors closely resembled that of primary tumors (Supplementary Fig. 7A). Tumors #10 and 12

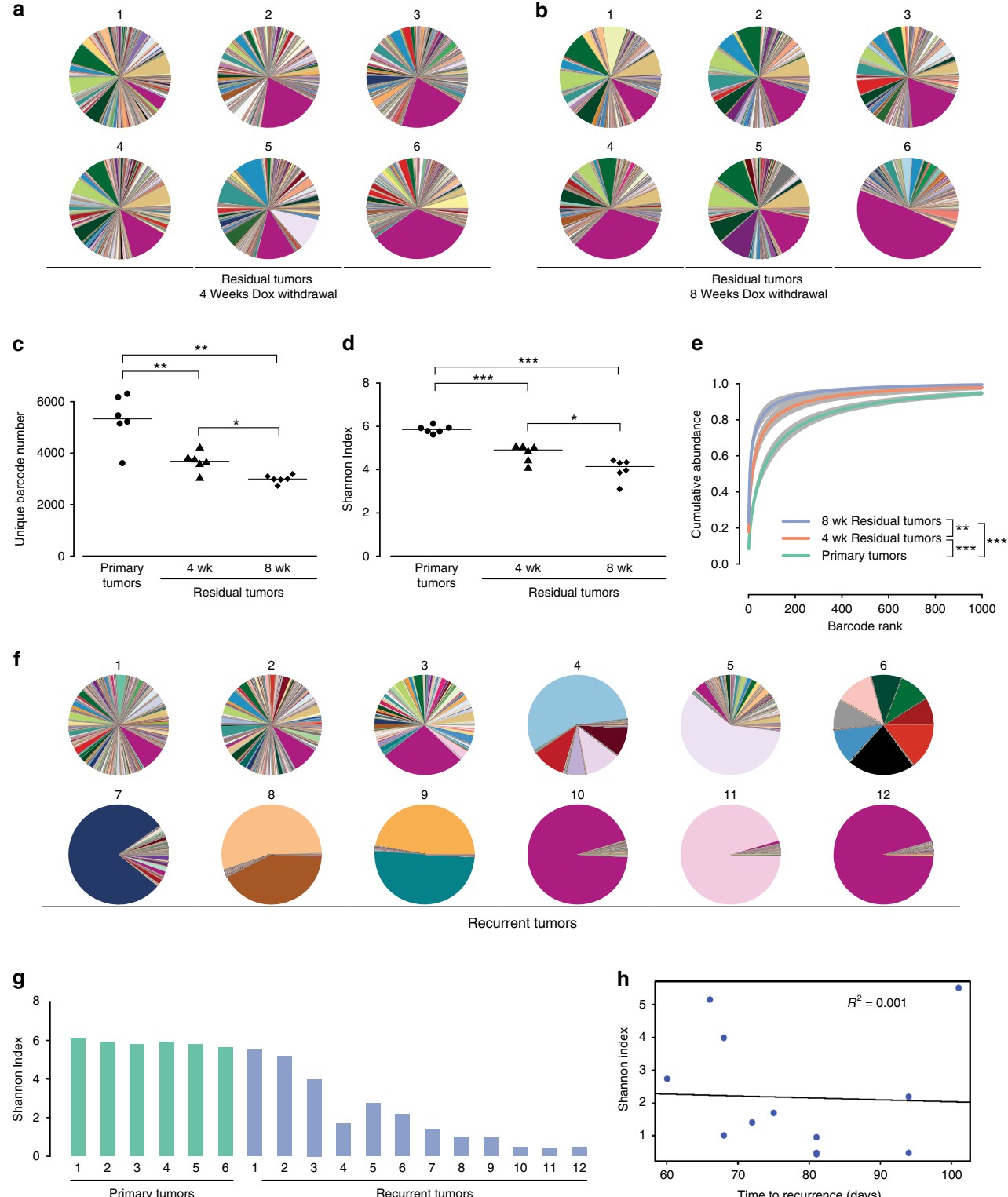

had intermediate similarity to primary tumors; this was influenced by the fact that barcode 8240:8518 was the most abundant barcode in these tumors (Supplementary Figs. 5A and 7A). In contrast, the remaining recurrent tumors were very dissimilar from primary tumors (Supplementary Fig. 7A), indicating that these tumors were dominated by distinct clones from those present in primary tumors.

Finally, we asked whether clonal and polyclonal tumors recurred with different kinetics. We found that there was no correlation between the clonal diversity of recurrent tumors and their time to recurrence (Fig. 3h). In sum, these results suggest that recurrence can proceed through at least two distinct routes. One route proceeds through a tumor-wide reactivation of all or most clones in a residual tumor, yielding a recurrent tumor with thousands of clones whose distribution resembles primary tumors. In the second route, only one or a few clones resume growth, giving rise to (oligo)clonal tumors dominated by clones distinct from those found in primary tumors.

**Fig. 3 Clonal complexity decreases during tumor regression, residual disease, and recurrence. a** Pie charts showing the relative frequency of barcodes in early residual tumors (4 weeks following Dox withdrawal). **b** Pie charts showing the relative frequency of barcodes in late residual tumors (8 weeks following Dox withdrawal). **c** Number of unique barcodes detected in primary tumors and residual tumors 4 and 8 weeks following Dox withdrawal. *$P <$ 0.05, **$P < 0.01$. Primary vs. 4-week residual tumors, $P = 2.6 \times 10^{-3}$; Primary vs. 8-week residual tumors, $P = 1.1 \times 10^{-3}$; 4-week residual tumors vs. 8-week residual tumors, $P = 0.024$. Significance determined by Welch Two-Sample $T$-test (two-sided). $n = 6$ biologically independent tumors in each cohort. **d** Shannon Diversity Index showing barcode complexity in primary tumors and residual tumors 4 and 8 weeks following Dox withdrawal. *$P < 0.05$, ***$P <$ 0.001. Primary vs. 4-week residual tumors, $P = 4.8 \times 10^{-4}$; Primary vs. 8-week residual tumors, $P = 1.1 \times 10^{-4}$; 4-week residual tumors vs. 8-week residual tumors, $P = 0.02$. Significance determined by Welch Two-Sample $T$-test (two-sided). $n = 6$ biologically independent tumors in each cohort. **e** Cumulative abundance plots for primary tumors and residual tumors 4 and 8 weeks following Dox withdrawal. Barcode complexity decreased following tumor regression and continued to decrease during the persistence of residual disease. **$P < 0.01$, ***$P < 0.001$. Primary vs. 4-week residual tumors, $P = 3.3 \times 10^{-4}$; Primary vs. 8-week residual tumors, $P = 2 \times 10^{-4}$; 4-week residual tumors vs. 8-week residual tumors, $P = 8 \times 10^{-3}$. Significance determined by Welch Two-Sample $T$-test (two-sided) between the number of barcodes composing 50% of reads. **f** Pie charts showing the relative frequency of barcodes in recurrent tumors. **g** Shannon Diversity Index showing barcode complexity in primary and recurrent tumors. $n = 6$ biologically independent primary tumors and $n = 12$ biologically independent recurrent tumors. **h** Barcode diversity in recurrent tumors is not correlated with time to recurrence.

---

***Met* amplification drives (oligo)clonal recurrences that are sensitive to Met inhibitors**. We next wanted to understand the mechanistic basis for the different clonal architecture found in recurrent tumors. Signaling through the receptor tyrosine kinase c-Met has been shown to be a common escape mechanism for tumors following loss of oncogenic signaling. For instance, *Met* amplification has been shown to promote resistance to EGFR inhibitors in EGFR mutant non-small cell lung cancer[17,32,33]. In a mammary tumor model with conditional PIK3CA expression, *Met* amplification drove tumor recurrence following PIK3CA downregulation[34]. Finally, increased signaling through Met was shown to promote recurrence in the same Her2-driven model we are using here[16]. We therefore tested whether *Met* amplification occurred in a subset of recurrent tumors, and whether differences in *Met* amplification status could underlie the different clonal composition of recurrent tumors. We measured *Met* copy number using a qPCR-based copy-number assay on genomic DNA, and found that six of the 12 recurrent tumors had *Met* amplification (Fig. 4a). Interestingly, the most abundant barcode(s) in each *Met*-amplified tumor was different (Fig. 4b), suggesting that these *Met*-amplified clones are distinct from one another. Consistent with this, mapping of the amplicon boundaries using qPCR on neighboring genes revealed that different recurrent tumors had distinct amplicons (Fig. 4c), suggesting that these *Met*-amplified tumors arose from different *Met*-amplified clones. Surprisingly, the barcodes marking these *Met*-amplified clones could be detected in all other recurrent tumors—as well as in all primary and residual tumors—but at much lower frequencies (Supplementary Fig. 7B–H). This suggests either that *Met* amplification is not by itself sufficient for recurrence, or that Met amplification occurs de novo in each tumor. In EGFR-mutant lung cancer, populations of cells that have preexisting resistant clones develop resistance more quickly than populations where resistance develops de novo[18]. We therefore compared the recurrence time between *Met*-amplified and non-amplified tumors. Surprisingly, we could detect no difference in recurrence-free survival between these tumors (Fig. 4d; $P$ value = 0.61, log-rank test).

We next determined whether *Met* amplification correlated with clonal diversity. All *Met*-amplified tumors had low clonal diversity (Fig. 4e), consistent with the finding that these tumors all had a small number of abundant clones (see Figs. 3f and 4b). *Met* was not amplified in any of the high-diversity tumors, or in any tumors whose clonal composition was highly correlated with primary tumors (Fig. 4e and Supplementary Fig. 7A). Taken together, these results suggest that distinct *Met*-amplified clones—possibly arising de novo—drive tumor recurrence in a subset of tumors, yielding (oligo)clonal recurrences with distinct clonal architecture from primary tumors.

We next tested whether *Met*-amplified tumors are sensitive to Met inhibitors. We injected the same starting population of barcoded cells (donor tumor #1) into a new cohort of recipient mice, and generated recurrent tumors as described above. We harvested and digested tumors to generate barcoded tumor cell cultures for in vitro analyses. We measured *Met* amplification and sequenced the barcodes to determine the clonal composition of these cultures. Cells cultured from recurrent tumor 1668 were not *Met*-amplified and were oligoclonal (Fig. 4f, g), while cells cultured from tumor 1669 were *Met*-amplified and clonal (Fig. 4f, g). We found that 1669 cells required HGF for their proliferation and were sensitive to the Met inhibitor crizotinib, while 1668 cells were insensitive to both HGF and crizotinib (Fig. 4h, i). To extend these results to an in vivo context, we established a recurrent cell line from a *Met*-amplified tumor arising in the autochthonous MTB;TAN model. This cell line was injected into the mammary fat pad of recipient mice, and mice were treated with vehicle or crizotinib. Crizotinib treatment significantly impaired the growth of these *Met*-amplified tumors (Supplementary Fig. 8A, B). Taken together, these results suggest, consistent with previous findings, that *Met*-amplified tumors are sensitive to Met inhibitors[32,35–38].

**Genomic characterization of recurrent tumors**. To identify other genetic alterations besides *Met* amplification that may drive recurrence, we performed whole-exome sequencing on a subset of the barcoded primary and recurrent tumors. We sequenced three primary tumors, five recurrent tumors with *Met* amplification, and five recurrent tumors without *Met* amplification. We also sequenced two recurrent tumor cell lines. Consistent with other reports, including a recent study examining Her2/neu-driven mammary tumors[39], we found that both primary and recurrent tumors had very few non-synonymous SNVs. Further, we did not identify any likely driver mutations in the top 20 most frequently mutated genes in human breast cancer, with the exception of *Pik3ca*, which was mutated in one of the recurrent tumor cell lines (Supplementary Table 1). In contrast, analysis of WES data revealed a number of copy-number alterations (CNAs) in both *Met*-amplified and non-amplified recurrent tumors (Supplementary Fig. 8C). While we could not identify any candidate driver oncogenes with focal, high-level amplification besides Met, this analysis does indicate that recurrent tumors have more CNAs as compared to primary tumors and suggests that the accumulation of CNAs is associated with tumor relapse (Supplementary Fig. 8C).

**Tumor recurrence is accompanied by an EMT**. To gain insight into the mechanism underlying reactivation in polyclonal recurrent tumors, we wanted to compare gene expression profiles of primary tumors, *Met*-amplified recurrent tumors, and

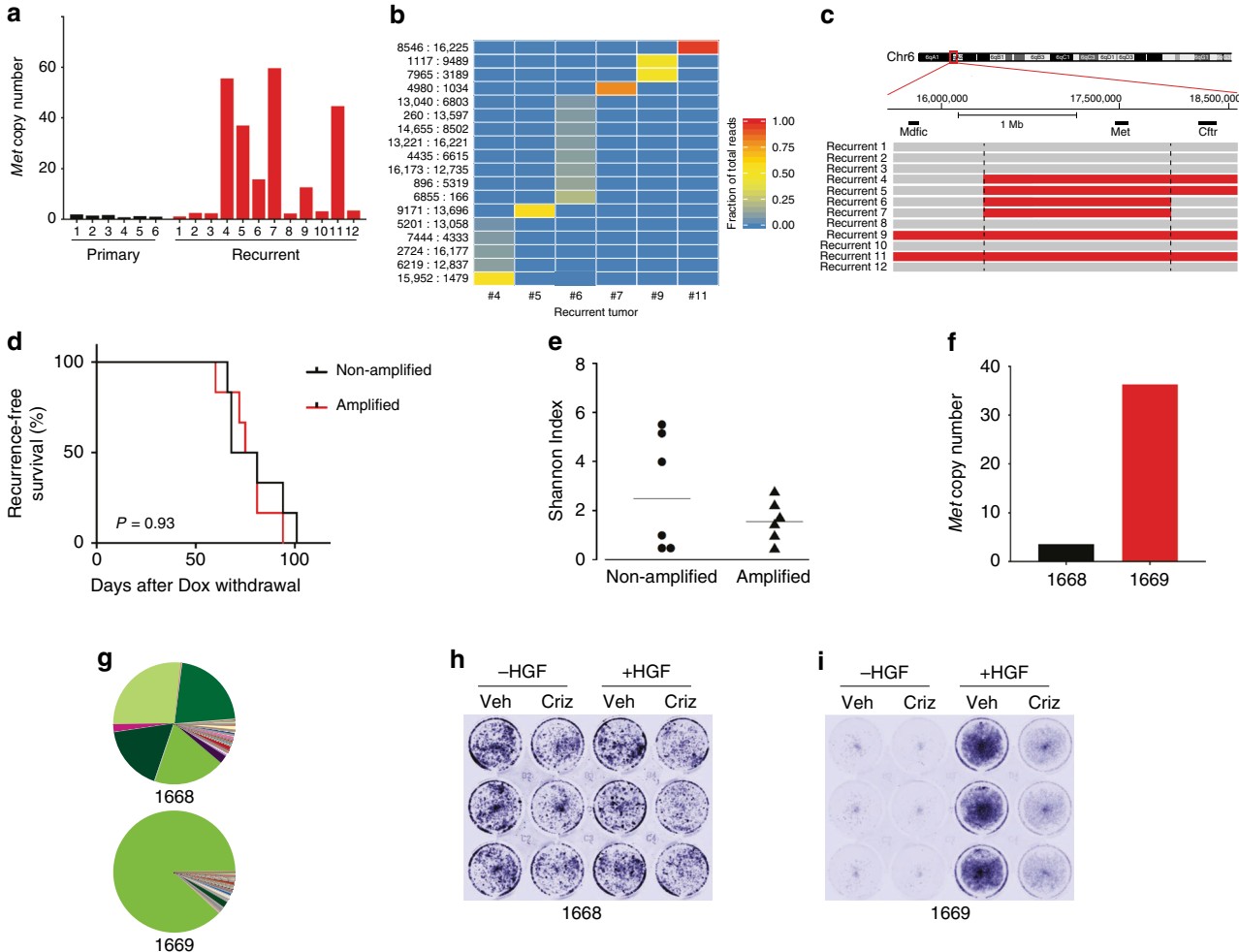

**Fig. 4 Met amplification drives recurrence in a subset of clonal recurrent tumors. a** qPCR analysis of Met copy-number in primary and recurrent tumors. Data are expressed as fold-increase in Met copy number relative to blood. $n = 6$ biologically independent primary tumors and $n = 12$ biologically independent recurrent tumors. **b** Heatmap of the abundance of selected barcodes in recurrent tumors, demonstrating that individual Met-amplified tumors have unique barcodes. All barcodes present at >5% in any tumor are shown. **c** Met-amplified recurrent tumors have distinct Met amplicons. qPCR analysis was used to measure Mdfic, Met, and Cftr copy number in individual Met-amplified tumors. Amplified regions are shown in red. **d** Kaplan–Meier recurrence-free survival curves for orthotopic tumors, stratified based on Met amplification status. Differences in survival were calculated using the Kaplan–Meier estimator with the log-rank (Mantel-Cox) test. $P = 0.637$. **e** Shannon Diversity Index showing barcode complexity in recurrent tumors with and without Met amplification. $n = 6$ biologically independent recurrent tumors without Met amplification and $n = 6$ biologically independent recurrent tumors with Met amplification. **f** Met copy number in two cell lines derived from orthotopic recurrent tumors with or without Met amplification. $n = 1$ cell line without Met amplification and $n = 1$ cell line with Met amplification. **g** Pie charts showing the relative frequency of barcodes in cells from **f**. **h–i** Met signaling drives tumor cell proliferation in Met-amplified recurrent tumor cells. Recurrent tumor cells 1668 (**h**) or 1669 (**i**) were grown in the presence of HGF and/or crizotinib for 3 days. Cells were stained using crystal violet.

polyclonal recurrent tumors. We could not perform gene expression analysis on the original cohort of tumors, since the entire tumor sample was used to isolate genomic DNA. Therefore, we generated a separate cohort of primary ($n = 4$) and recurrent barcoded ($n = 8$) tumors using a second digested MTB; TAN tumor (donor tumor #2), and isolated both DNA and RNA from these tumors. Barcode sequencing revealed that the pattern of changes in the clonal composition of these tumors was similar to the original cohort, with a progressive decrease in the complexity of barcodes during recurrence (Supplementary Fig. 9A–C). Similarly, half ($n = 4$) of the recurrent tumors (recurrent tumors #1, 2, 3, and 6) had Met amplification (Supplementary Fig. 8D), and these tumors were either clonal or oligoclonal (Supplementary Fig. 4B). In contrast, tumors lacking Met amplification (recurrent tumors 4, 5, 7, and 8) were predominantly polyclonal (Supplementary Fig. 9B).

Using this new cohort, we examined how gene expression patterns differed between primary tumors, Met-amplified recurrent tumors, and non-Met-amplified polyclonal recurrent tumors. It has previously been shown that recurrent tumors arising in MTB;TAN mice have undergone an EMT, and experimental induction of EMT by expression of the transcription factor Snail accelerates recurrence in this model[14]. Consistent with this, EMT can promote resistance to both targeted and chemotherapies[40,41]. We therefore determined whether clonal Met-amplified recurrent tumors and/or polyclonal non-Met-amplified recurrent tumors had gene expression changes suggestive of EMT. We found that all recurrent tumors had downregulated expression of the epithelial markers E-cadherin (Cdh1) and Epcam, and upregulated the mesenchymal marker Ddr2, as compared to primary tumors (Fig. 5a and Supplementary Fig. 10A); these genes have been proposed as key EMT markers in vivo[42]. Similarly, E-cadherin (Cdh1) and Epcam expression were

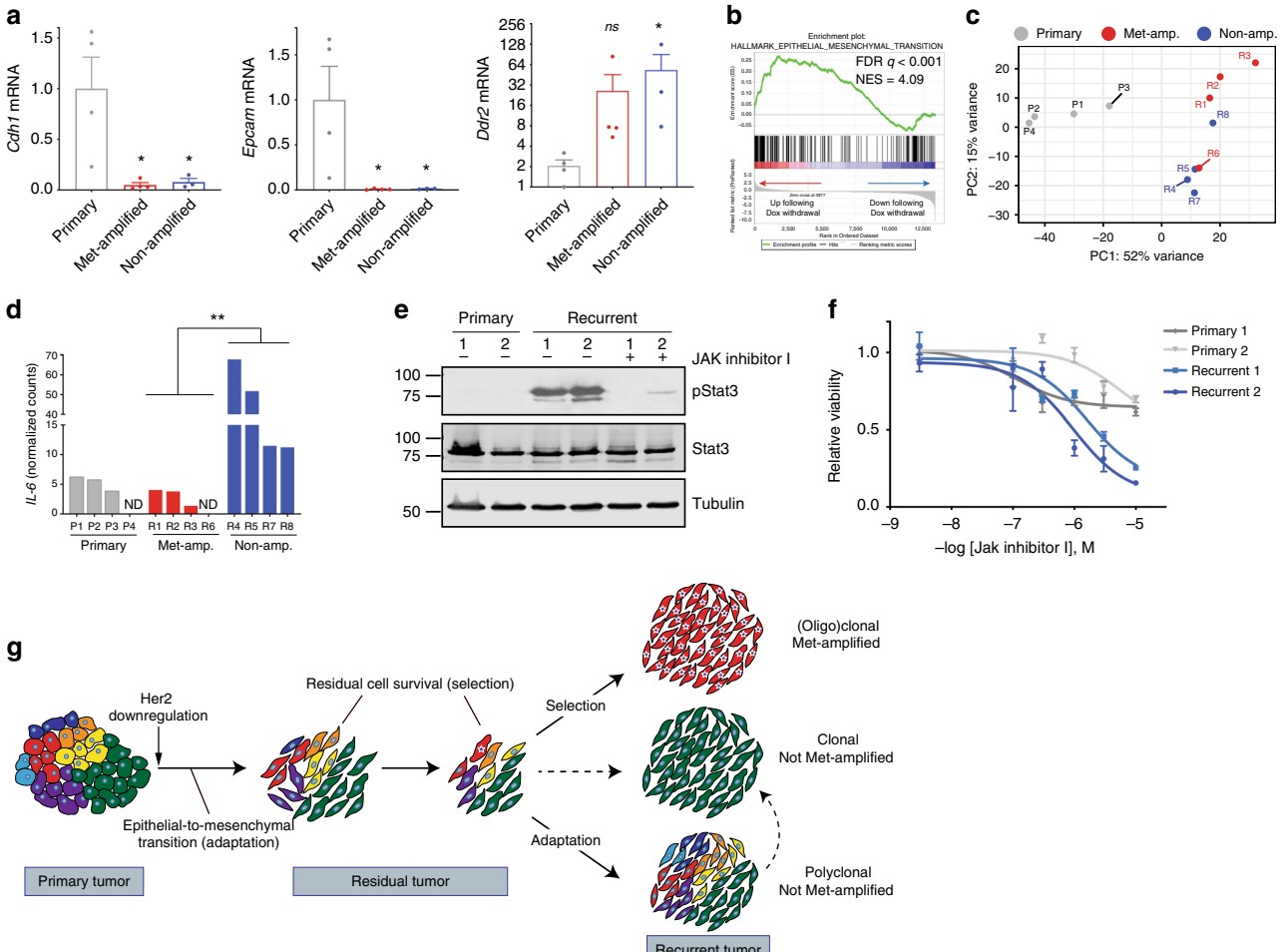

**Fig. 5 Polyclonal recurrent tumors have distinct gene expression profiles. a** qRT-PCR analysis showing expression of epithelial (Cdh1 and Epcam) and mesenchymal (Ddr2) markers in primary tumors, Met-amplified recurrent tumors, and non-amplified recurrent tumors. Significance was determined using one-way ANOVA with Dunnett's multiple comparison testing. *$P < 0.05$ between each recurrent cohort and the primary tumor cohort. Cdh1: primary vs met-amplified, $P = 0.015$; primary vs. non-amplified, $P = 0.025$. Epcam: primary vs met-amplified, $P = 0.027$; primary vs. non-amplified, $P = 0.039$. Ddr2: primary vs met-amplified, $P = 0.11$; primary vs. non-amplified, $P = 0.01$. $n = 4$ biologically independent primary tumors, $n = 4$ biologically independent Met-amplified recurrent tumors, and $n = 3$ biologically independent non-Met-amplified recurrent tumors in each cohort. Data are presented as mean ± SEM. **b** Gene set enrichment analysis showing enrichment of an EMT signature in cells following Her2 downregulation in vitro. Normalized enrichment score was calculated using the Kolmogorov–Smirnov statistic. To correct for multiple testing, the FDR q-value was estimated using permutation testing to compare the actual NES to random gene sets. **c** Principal components analysis (PCA) of gene expression profiles in primary tumors, and recurrent tumors with and without Met amplification. **d** Normalized counts of IL-6 expression from RNA-seq analysis of primary tumors, Met-amplified recurrent tumors, and non-amplified recurrent tumors. Significance was determined using one-way ANOVA with Tukey's multiple comparison testing. ANOVA $P = 0.039$. *$P < 0.05$ between Met-amplified recurrent tumors and non-amplified recurrent tumors (Met-amplified vs. Non-amplified, $P = 0.048$). $n = 4$ biologically independent primary tumors, $n = 4$ biologically independent Met-amplified recurrent tumors, and $n = 4$ biologically independent non-Met-amplified recurrent tumors in each cohort. **e** Western blot showing Stat3 phosphorylation (Y705) in primary and recurrent tumor cells derived from the autochthonous model. Note that the primary cells are donor tumor #1 and 2, and both recurrent tumor cells do not have Met amplification. Molecular weight markers (kDa) are shown at the left. Results are representative of three independent experiments. **f** Dose-response curves of primary and recurrent tumor cells treated with increasing concentrations of Jak inhibitor I. $n = 2$ biologically independent primary tumor cell lines and $n = 2$ biologically independent recurrent tumor cell lines examined over three independent experiments. Data are presented as mean ± SEM. **g** Model showing adaptation and selection shaping the clonal composition of tumors during residual disease and recurrence.

downregulated, and the mesenchymal genes Vimentin (*Vim*) and *Ddr2* were upregulated in both 1668 and 1669 recurrent tumors cells as compared to cells derived from primary donor tumors (Supplementary Fig. 10B–E). These results indicate that all recurrent tumors had undergone EMT, irrespective of their clonality or *Met* amplification status.

We reasoned that two mechanisms could explain this finding: recurrence could select for preexisting mesenchymal cells in primary tumors, or EMT could be an adaptive response to Her2 inhibition. The finding that even polyclonal recurrent tumors had

undergone EMT argued against a selection for preexisting clones, since the clonal composition of polyclonal recurrent tumors resembled primary tumors. To directly assess whether EMT is an adaptive response to Her2 inhibition, we cultured cells derived from a primary tumor and removed dox to induce Her2 downregulation in vitro. We then performed RNA-seq to identify transcriptional changes 4 days following Her2 downregulation, and performed gene set enrichment analysis (GSEA) to identify gene sets enriched in cells with and without Her2 expression. The top gene set enriched in cells grown in the presence of dox was an

E2F target gene set (Supplementary Fig. 10F and Supplementary Data 3), consistent with the notion that Her2 drives proliferation of these cells[15], and thereby providing validation of this approach. We found that the top-scoring gene set enriched in the cells following dox withdrawal was an EMT signature (Fig. 5b and Supplementary Data 4). This is consistent with the model that EMT is an adaptive response to Her2 downregulation that occurs in the absence of selection, providing a basis for the observation that all recurrent tumors have undergone EMT.

**Polyclonal recurrent tumors have distinct gene expression profiles.** We next performed RNA-seq on tumors from this cohort to more broadly compare gene expression patterns between primary tumors, *Met*-amplified recurrent tumors, and non-*Met*-amplified polyclonal recurrent tumors (Supplementary Data 5 and 6). Principal components analysis revealed that these tumors clustered by group, indicating that each of these cohorts had a distinct gene expression pattern (Fig. 5c). Principal component 1 (PC1) efficiently separated primary tumors from recurrent tumors, while PC2 partly separated *Met*-amplified from non-amplified tumors (Fig. 5d). Taken together, this suggests that polyclonal recurrent tumors have a unique gene expression profile.

The polyclonal nature of these tumors suggested that recurrence was associated with reactivation of a large number of clones in residual tumors. We reasoned that this may have been driven by a secreted factor(s) that acted in an autocrine or paracrine manner to induce tumor cell proliferation. We therefore examined the RNA-seq data for cytokines expressed at higher levels in non-*Met*-amplified recurrent tumors. All cytokines from the KEGG_CYTOKINE_CYTOKINE_RECEP-TOR_INTERACTION gene list were ranked based upon their fold-change in expression between non-amplified recurrent tumors and Met-amplified recurrent tumors. We found that IL-6 was the second most differentially expressed cytokine, and was expressed between 4.9-fold and 29-fold higher in non-amplified recurrent tumors compared to *Met*-amplified recurrent tumors (Fig. 5d; $P$ value = 0.006, ANOVA with Tukey's multiple comparison testing). IL-6 exerts its effects by signaling through Jak kinases and Stat transcription factors[43]. We therefore examined whether there was evidence for Jak-Stat pathway activation in recurrent tumors lacking *Met* amplification. Because we did not have protein lysates available for this cohort of tumors, we examined tumor cells cultured from *Met*-amplified or non-amplified autochthonous recurrent tumors arising in MTB;TAN mice. We found that levels of pStat3 were elevated in cells from non-amplified recurrent tumors, and this could be inhibited by a Jak kinase inhibitor (Fig. 5e). Finally, we tested whether tumor cells from non-amplified recurrent tumors were sensitive to a Jak inhibitor. Treatment with a pan-Jak kinase inhibitor had a modest effect on primary cells, but nearly completely inhibited the growth of recurrent tumor cells (Fig. 5f). These results suggest that polyclonal recurrent tumors are driven in part by autocrine or paracrine IL-6 production that signals through the Jak-Stat pathway to drive the proliferation of these cells.

## Discussion

Most deaths from breast cancer are caused by the survival and eventual recurrence of residual tumor cells after therapy[7]. Understanding the pathways that govern residual cell survival, and how these cells resume proliferation, may suggest strategies for targeted approaches to prevent or delay relapse. As a first step in this direction, it is important to understand the clonal evolution of tumors during residual disease and recurrence; that is, to what extent does the clonal composition of residual and recurrent tumors resemble that of primary tumors? While next-generation sequencing has provided insights into clonal evolution in recurrent tumors in humans[44], the difficulty in identifying and sampling residual tumors has impeded progress in studying this critical stage. In the current study, we used DNA barcoding to directly monitor clonal evolution during tumor growth, regression, residual disease, and recurrence.

The clonal composition of recurrent tumors suggested that recurrence can proceed through several distinct routes (Fig. 5g). One subset of recurrent tumors was clonal or oligoclonal, exhibited amplification of *Met*, and was sensitive to Met inhibitors. In these tumors, recurrence is likely driven by the expansion of a *Met*-amplified clone. Interestingly, the finding that each *Met*-amplified tumor had a unique barcode and distinct amplicon boundaries suggests that these tumors arose from independent *Met*-amplified clones. This may argue against the presence of a single preexisting clone with *Met* amplification, and instead suggests that amplification of *Met* occurs de novo, either during orthotopic tumor growth or at the residual disease stage. It is important to note, however, that we cannot rule out the possibility that the donor primary tumor had many independent *Met*-amplified clones, each of which was labeled with a different barcode. A recent study by Seth et al. devised an elegant approach for prospectively isolating clones of interest from a complex population of barcodes[28]. We identified a number of clones in recurrent tumors that have *Met* amplification, and the barcodes marking these clones are known (Fig. 4b). By isolating these barcoded clones from the injected cell population and measuring *Met* copy number, we could definitively determine whether these clones harbor preexisting *Met* amplification. We are currently working to implement this approach. Notwithstanding the timing of *Met* amplification, these results demonstrate that *Met* amplification is an escape route for tumors following oncogene inhibition, consistent with findings from a number of other groups[32–35,41].

A second subset of tumors was polyclonal; these tumors were composed of many hundreds of clones, and the distribution of these clones was similar to primary tumors. These tumors lacked *Met* amplification and were not sensitive to Met inhibitors. In these tumors, recurrence was likely driven by the tumor-wide reactivation of all or most of the clones present in residual tumors, resulting in recurrent tumors with a clonal architecture that resembled primary tumors. It is possible that in these tumors reactivation of residual tumor cells was mediated by a secreted factor that functions in a paracrine manner, acting on all clones. Consistent with this, polyclonal recurrences expressed high levels of IL-6 and Stat3 activation, and were sensitive to a Jak inhibitor. Interestingly, an IL-6-driven paracrine loop has been shown to drive the survival of residual lymphoma cells following chemotherapy in the thymus[45]. Our results extend this finding by suggesting that, in some cases, tumor cell-derived IL-6 may act in an autocrine manner to drive the proliferation of residual tumor cells.

In the final group of tumors, a single clone marked by barcode 8240:8518 composed nearly the entire tumor. This suggests that this clone, which was the most abundant clone present in all primary and residual tumors, expanded to give rise to recurrent tumors. These tumors were not *Met*-amplified, suggesting that an alternative pathway drives relapse in these tumors. It will be interesting to elucidate the event that triggers reactivation of this clone, which may include acquisition of CNAs (Supplementary Fig. 8).

One striking finding from these studies is the apparent diversity of recurrence paths accessible to tumors. While *Met*-amplified recurrent tumors had similar gene expression patterns to one another, recurrent tumors lacking gene *Met* amplification were

quite heterogeneous with respect to both clonal composition and gene expression profiles. These results are similar to studies analyzing clonal architecture in TNBC metastasis, which observed substantial variations in the clonal complexity of metastases as compared to corresponding primary tumors[23,26]. Future work will be needed to define the common phenotypic properties shared by tumors with distinct clonal architecture but similar propensities to recur.

Her2 downregulation and tumor regression were accompanied by a progressive decrease in the clonal diversity of tumors, consistent with the notion that only a subset of tumor cells is capable of surviving oncogene downregulation. Interestingly, we found that the residual disease stage itself is also accompanied by a decrease in the number of clones and their diversity within tumors. This suggests that, rather than being a static state, residual disease is a dynamic process with ongoing changes in clonal composition. Indeed, the reduction in diversity between early and late residual disease was as large as the reduction that accompanied tumor regression, suggesting that the magnitude of selective pressures during the residual disease stage was nearly equivalent to the selective pressure exerted by oncogene inhibition. While the cellular stresses underlying this selection remain unknown, this finding suggests that recurrent tumors may arise from a subset of residual cells that can overcome these stresses. Our findings are similar to a study that assessed clonal heterogeneity of disseminated residual tumor cells in humans[46]. In this study, the authors showed that residual tumors comprise a heterogeneous mixture of clones, while clinically evident metastases seemed to result from the expansion of a subset of these clones. This suggests that the clonal changes that accompany local residual disease and recurrence in our mouse model mirror, at least in part, what is observed in disseminated residual disease and metastatic relapse in humans.

Changes in the clonal composition of tumors at different stages of recurrence suggested that both adaptive and selective pressures act to drive tumor evolution and relapse. All tumors—irrespective of their clonality or Met amplification status—had gene expression patterns consistent with EMT. This suggests that EMT did not result from the selection of a subset of clones with mesenchymal characteristics, since the clonal composition of polyclonal recurrent tumors was very similar to primary tumors. Instead, these results indicate that EMT may be an adaptive response to oncogene inhibition. Gene expression changes in tumor cells shortly following Her2 downregulation are consistent with this. It is important to note, however, that experimental induction of EMT accelerates recurrence in this model[14]. Taken together, these findings suggest that EMT is an adaptive response that promotes cell survival following oncogene inhibition, but that the ability to undergo a full EMT is nonetheless rate-limiting for recurrence, perhaps because mesenchymal cells can better survive oncogene downregulation. Consistent with this, residual breast cancer cells that can survive neoadjuvant therapy display mesenchymal characteristics[47]. More broadly, the observation that some recurrent tumors had very similar clonal composition to primary tumors is prima facie evidence that recurrence in this subset of tumors was driven by adaptive mechanisms. In contrast, Met-amplified recurrent tumors usually comprised one or two clones, suggesting that recurrence in these tumors proceeds via selection. Together, these results indicate that adaptation and selection can work together to shape the evolution of tumors during residual disease and recurrence.

The findings presented here are reminiscent of results describing the emergence of resistance to EGFR inhibitors in EGFR-mutant lung cancer[18] and relapse following chemotherapy in triple-negative breast cancer[22]. Hata et al. found that resistance could emerge either from the expansion of cells with preexisting resistance mutations, or through the survival of a population of

drug-tolerant persister cells[18]. In this latter case, drug-tolerant cells can survive but not proliferate in the presence of EGFR inhibitors; these cells eventually evolve mutations rendering them fully resistant. Similarly, Echeverria et al. found that residual and recurrent TNBCs following chemotherapy have similar clonal composition to untreated tumors[22]. In our study, we find that residual cells are capable of surviving Her2 downregulation but are not initially competent to proliferate, suggesting that these cells may be analogous to persister cells. A unique finding in our study is that, in some instances, the majority of these residual cells can acquire the ability to resume proliferation in the absence of additional clonal selection. This represents a mode of recurrence in which some signal—perhaps a secreted paracrine-acting factor such as IL-6—can induce the reactivation of residual tumor cells en masse. Approaches to target this resistance mechanisms may block this route of recurrence. More broadly, the observation that tumor recurrence can arise through a polyclonal, drug-tolerant intermediate stage seems to be common across different tumor types and therapies.

We examined the clonal dynamics of tumor regression, residual disease, and recurrence locally in the mammary gland. While locoregional recurrences represent important clinical problems in themselves, most deaths from breast cancer result from distant relapse. Several lines of evidence suggest that the survival and recurrence of cancer cells locally in the breast are mechanistically linked to the survival and recurrence of disseminated cells. First, extensive evidence has shown that the response of tumors locally to neoadjuvant therapy is a strong predictor of eventual recurrence at distant sites, even when the tumor in the breast is surgically excised[48]. Second, both the incidence and timing of local recurrence are strongly correlated with distant relapse[49,50]. These associations suggest that studying the process of local residual disease and relapse can provide insight into disseminated residual disease and metastatic recurrences. However, there are undoubtedly important differences between local and distant relapse, including the bottleneck of dissemination itself and stresses imposed by the foreign microenvironment. A number of studies have used DNA barcoding to study the clonal dynamics of metastatic spread; extending these studies to examine how disseminated tumors respond to therapy and develop resistance remains an important goal.

## Methods

**Cell culture, barcode library transduction, and dose-response curves**. Primary tumor cells from two independent MTB;TAN primary tumors were used for all experiments[11]. These cells were cultured in the presence of dox to maintain Her2 expression and used at early passage number (prior to passage 15). To generate barcoded populations, cells were infected with the CellTracker lentiviral barcode library from Cellecta, which contains ~50 million unique barcodes. Overall, 200,000 cells were transduced with this library in the presence of polybrene (8 µg/ml) at an MOI of 0.1 to yield a population with ~20,000 unique barcodes. Cells were expanded for 12 population doublings prior to injection into mice.

Cell lines were generated from barcoded recurrent tumors using enzymatic digestion[11]. Briefly, tumors were harvested, minced, and digested in collagenase and hyaluronidase (StemCell Technologies). Digestion media was removed and the cells were resuspended in red blood cell lysis buffer, followed by Dispase II (5 mg/mL) and DNase I (100 µg/ml) prior to plating. Cells were grown in DMEM with 10% SCS, 1% Pen-Step, 1% Glutamine, and supplemented with EGF (0.01 µg/ml, Sigma) and insulin (5 µg/ml, Gemini Bioproducts). To generate Met-amplified cell cultures, cells were supplemented with HGF (250 ng/ml, R&D systems). Crizotinib was used at 500 nM (Selleck). Recurrent tumor cells derived from recurrent tumors arising in the autochthonous MTB;TAN mouse model were cultured in the absence of dox[11]. All cell lines were tested for mycoplasma contamination by the Duke Cell Culture Facility and tested negative. A summary of the cell lines and tumors used in this study is shown in Supplementary Table 2.

Dose-response curves were performed using CellTiterGlo[11]. Briefly, 1000 primary or recurrent tumor cells were plated onto 96-well plates. The following day, media containing 1% serum with increasing concentrations of Jak Inhibitor I (Calbiochem) was added to cells. Cell viability was measured 3 days later using CellTiterGlo (Promega) on a Biotek Synergy HTX plate reader with Gen5 software version 2.06.

For RNA-seq analysis, primary Her2-driven tumor cells were cultured as mammospheres[51] in the presence of 2 µg/ml dox. To induce Her2 downregulation, dox was removed from cultures, and RNA was harvested 4 days later.

**Mice**. All experiments were approved by Duke IACUC (Approval #A199-17-08). Mice were maintained on 12-hr light/12-hr dark schedule, at a temperature (20–26 °C) and relative humidity (40–70%) range recommended by *The Guide for the Care and Use of Laboratory Animals*. One million barcoded MTB;TAN tumor cells were injected into the inguinal mammary gland of nu/nu mice. Mice were provided dox (2 mg/ml, 5% sucrose) in their drinking water 2 days prior to injection and remained on dox during the course of primary tumor growth. Mice were palpated twice per week to monitor growth. Once tumors reached 5 mm in diameter, one cohort of mice was sacrificed, and dox was removed from the drinking water of the other cohorts to initiate Her2 downregulation. Additional cohorts were sacrificed at 4 weeks and 8 weeks following dox withdrawal. A final cohort was monitored for the appearance of recurrent tumors, and mice were sacrificed when tumors reached between 8 and 12 mm in diameter. Primary and recurrent tumors were harvested from mice and snap frozen for DNA isolation. Residual tumors were microdissected with the aid of a fluorescent microscope, and ample margins (2 mm) were included to ensure that no residual tumor cells were missed during dissection.

For metastasis experiments, 6-week-old-female nu/nu mice were provided dox (2 mg/ml, 5% sucrose) in their drinking water 2 days prior to injection. Overall, 100,000 tumor cells were injected into the tail vein of mice in 50 µl of PBS. Mice were injected with D-luciferin (150 mg/kg, PerkinElmer) 10 min prior to imaging and then imaged on an IVIS Lumina III bioluminescence/fluorescence imager (PerkinElmer). Data were analyzed using LivingImage 4.7.3 (PerkinElmer).

To determine the effect of a Met inhibitor on the growth of Met-amplified tumors in vivo, nude mice were injected with the Met-amplified recurrent tumor cell line (recurrent #3). Overall, $1 \times 10^6$ cells were injected into the inguinal mammary gland of nu/nu mice. One tumors were palpable, mice were randomized to receive vehicle control or crizotinib (50 mg/kg) by oral gavage daily. Tumor growth was measured twice weekly using calipers.

**DNA isolation, library preparation, and barcode sequencing**. To isolate genomic DNA, tumors were digested with Proteinase K in Buffer ATL (Qiagen). Following digestion, cells were lysed by adding SDS (final concentration, 0.5%) and sonicating to shear DNA. Genomic DNA was then purified using phenol–chloroform extraction followed by isopropanol precipitation. DNA yield from all samples was >10 µg (minimum $1.5 \times 10^6$ cells). This genomic DNA was used for barcode PCR, qPCR, and whole-exome sequencing.

To prepare libraries for sequencing, DNA concentration was measured using Broad Range Qubit methodology (Invitrogen). In total, 80 ng of DNA (~12,100 genomes) was used as input for each single reaction. To increase the number of analyzed cells we performed 8 reactions per sample; therefore, barcodes from around 97,000 cells were analyzed for each sample. We used a two-step PCR amplification protocol as described in Lundberg et al.[52] In the first step of the reaction barcodes were amplified using custom primers (Forward: Step1_F, Reverse: Step1_R) for 10 cycles. KAPA Robust 2G polymerase (KAPA Biosystems-Roche) was used during this step to ensure efficiency and specificity of barcode amplification. DNA was purified between steps one and two using magnetic beads at a 1.3(beads):1(reaction) proportion. During the second step, PCR products from step one were amplified using custom primers (Forward: ClonalBarcode Adaptor_1, Reverse: PCR Primer_IndexN) for 23 cycles. KAPA Hifi Polymerase (KAPA Biosystems-Roche) was used during this step to ensure fidelity of the amplification. DNA was purified after the second step using magnetic beads at a 1(beads):1(reaction) proportion. Amplified barcodes were sequenced using custom Clonal Barcodes R1 sequencing primer and standard TruSeq sequencing primers using an Illumina HiSeq2500 Rapid Single End 40 × 9. Custom primer sequences are shown in Supplementary Table 3.

**RNA-seq, whole-exome sequencing, qRT-PCR, copy-number analysis, and western blotting**. RNA was isolated from cells or tumors using the RNeasy kit (Qiagen). RNA was sequenced using the Illumina HiSeq 4000 libraries and sequencing platform with 50 base pair (bp) single end reads by the Duke GCB Sequencing and Genomic Technologies Shared Resource (Durham, NC).

Whole-exome sequencing was performed using the Agilent SureSelect XT Mouse All Exon Capture Kit and sequenced using the Illumina HiSeq 4000 with 150 bp paired-end reads by the Duke GCB Sequencing and Genomic Technologies Shared Resource (Durham, NC).

qRT-PCR was performed as described[11] using the following TaqMan probes: Cdh1 (Mm01247357_m1), Epcam (Mm00493214_m1), Ddr2 (Mm00445615_m1), and Vim (Mm01333430_m1) and normalized to expression of Actb (Mm02619580_g1).

To measure the copy number of Met and flanking genes, we performed real-time PCR on purified genomic DNA using the following TaqMan probes (all from Applied Biosystems): Met (Mm00565151_cn), Mdfic (Mm00565127_cn), Cav2 (Mm00181583_cn), and Cftr (Mm00181608_cn). Real-time PCR was performed on a Bio-Rad CFX384 Real-Time PCR system and acquired with Bio-Rad CFX Manager Software Version 3.1. Absolute copy-number for each gene was calculated

by normalizing to an internal reference gene (Tfrc) and expressed as fold-change over genomic DNA from nontumor tissue.

Western blotting and immunofluorescence were performed[53] using antibodies against total Stat3 (Cell Signaling, 1:1000 for western blotting), phospho-Stat3 Y705 (Cell Signaling, 1:1000 for western blotting), E-cadherin (Cell Signaling, 1:100 for immunofluorescence), and tubulin (Sigma, 1:1000 for western blotting). Uncropped western blots are shown in Source Data File. Immunofluorescence images were captured on a Zeiss AxioImager Z2 microscope with Zeiss ZEN 2 software Version 3.2.

**Single-cell RNA-sequencing**. Libraries were prepared following the 10× Genomics Single-Cell protocols. Briefly, single cells were dissociated, then washed and resuspended in a 1× PBS/0.04% BSA solution, at a concentration of 1000 cells/µl. After selection (<50 µm), the cell suspension was washed with a 1× PBS/0.04% BSA solution to remove debris, clumps, dead cells and contaminants, and a Cell-ometer (Nexcelom, Lawrence, MA) used to determine the cell viability and concentration to normalize to $1 \times 10^6$ cells/ml. Samples were titered to contain ~5000 cells per library construction. Cells were then combined with a master mix that contains reverse-transcription reagents. The gel beads carrying the Illumina TruSeq Read 1 sequencing primer, a 16 bp 10× barcode, a 12 bp unique molecular identifier (UMI) and a poly-dT primer were loaded onto the chip together with oil for the emulsion reaction. The Chromium Controller partitions the cells into nanoliter-scale gel beads in emulsion (GEMS) within which reverse-transcription occurs. All cDNAs within a GEM, i.e. from one cell, share a common barcode. After the RT reaction, the GEMs were broken and the full length cDNAs cleaned with both Silane Dynabeads and SPRI beads. After purification, the cDNAs were assayed on an Agilent 4200 TapeStation High Sensitivity D5000 ScreenTape (Santa Clara, CA) for qualitative and quantitative analysis.

Enzymatic fragmentation and size selection were used to optimize the cDNA amplicon size. Illumina P5 and P7 sequences (San Diego), a sample index, and TruSeq read 2 primer sequence were added via End Repair, A-tailing, Adapter Ligation, and PCR. The final libraries contained P5 and P7 primers used in Illumina bridge amplification. Sequence was generated using paired-end sequencing (one end to generate cell specific, barcoded sequence and the other to generate sequence of the expressed poly-A tailed mRNA) on an Illumina sequencing platform at a minimum of 50k reads/cell.

**Single-cell sequencing data analysis**. The primary analytical pipeline for the SC analysis follows the recommended protocols from 10X Genomics. Briefly, raw base call (BCL) files generated by Illumina sequencers were demultiplexed into FASTQ files, and then alignment to the mouse reference transcriptome, filtering, barcode counting, and UMI counting were performed using the most current version of 10X's Cell Ranger software (version 3.1.0). Chromium cell barcodes were used to generate feature-barcode matrices encompassing all cells captured in each library. The secondary statistical analysis was performed using the R package Seurat[54], which performs quality control and subsequent analyses on the feature-barcode matrices produced by Cell Ranger. In Seurat, data were first normalized and scaled after basic filtering for minimum gene and cell observance frequency cutoffs, and further filtering was performed to identify and exclude possible multiplets (i.e. instances where more than one cell was present and sequenced in a single emulsified gel bead). The removal of further technical artifacts was performed using regression methods to reduce noise. After quality control procedures, we performed linear dimensional reduction calculating principal components using the most variably expressed genes in the dataset. Cells were grouped into an optimal number of clusters for de novo cell type discovery using Seurat's FindNeighbors() and FindClusters() functions, graph-based clustering approaches with visualization of cells was achieved through the use of manifold learning technique UMAP (Uniform Manifold Approximation and Projection), which reduces the information captured in the selected significant principal components to two dimensions[55].

**Bioinformatics**. For barcode mapping, low quality reads with mean Phred scores smaller than 30 were excluded from the analysis. Remaining reads with a length of 40 bp were split into left and right short reads (18 bp) after removing the 4 bp linkers at the center. The resulted short left and right RNA reads were mapped to the sequencing library of 50 million unique barcodes. The mapping allowed up to two mismatches. After mapping, the mapped results for short left and right RNA reads were combined. Based on the combined mapping results, a count table listing the number of reads for each barcode in each sample was generated for the further statistical analysis. Barcodes with read counts smaller than 1 were excluded from downstream statistical analysis. Numbers of unique barcodes, proportions of reads for the barcodes, most abundant sequenced barcodes, and Shannon diversity indexes for each sample were obtained based on the barcode reads count table. Dissimilarities based on Jensen–Shannon divergence were obtained to evaluate the differences in clonal composition among different tumors.

For analysis of RNA-seq data, FastQC (v0.11.5) (Andrews, S., FastQC A Quality Control tool for High Throughput Sequence Data. 2014.; https://www.bioinformatics.babraham.ac.uk/projects/fastqc/) was used to assess the general sequencing quality. All the samples were sequenced at a length of 51 bp. All the samples have an averaged Phred score >30. RNA sequences which passed quality

control (mean Phred score >10) were aligned to reference mouse genome GRCm38 (mm10) using STAR (v2.5.2b)[56]. The alignment report is shown in Supplementary Table 4. The aligned reads were mapped to genomic features using HTSeq[57], implemented in the STAR program. The reference sequence and GTF file were obtained from the NCBI GRCm38 bundle available from the iGenomes collection. Gene differential expression was performed within the framework of a negative binomial model using R (v3.4.4) (R Core Team, R: A Language and Environment for Statistical Computing. 2016: Vienna, Austria.) and its extension package DESeq2 (v1.18.1)[58]. PCA analysis were performed for each of the cohort based on the normalized reads matrix (based on rlog from DESeq2(v1.18.1)) from the DESeq2 (v1.18.1).

For whole-exome sequencing analysis, single-nucleotide variants (SNVs) and CNAs were detected using standard approaches[59]. Briefly, Mutect2[60] was used to call non-synonymous SNVs by comparing whole-exome sequencing data against matched normal DNA, and CNAs were identified using CODEX2[61].

**Statistics**. A t-test was used to test the changes of the number of unique barcodes, the Shannon diversity index, the number of the most abundant barcodes that composing 50% of the total reads, and the most abundant barcode proportion for injected cells, primary tumors, and residual tumors. t-test P values were reported. Two-way ANOVA test was used to test the mean changes of Shannon diversity indexes from primary tumors to 4-week residual tumors and 4-week residual tumors to 8-week residual tumors. Unadjusted P value of the interaction term from two-way ANOVA test was reported. Cox proportional hazard regression was used for the time to tumor recurrence analyses[62] using the R extension package survival v2.41-3 (Therneau T., A Package for Survival Analysis in S. version 2.38, 2015; https://CRAN.R-project.org/package=survival). The Cox score statistics[62] was used to test the difference of time to tumor recurrence between Met-amplified and non-Met-amplified tumors. The P values were not adjusted for multiple testing. The Kaplan–Meier product estimator was used to estimate the survival function for the tumor recurrence for both Met-amplified and non-Met-amplified recurrent tumors. Graphs were created in R or using GraphPad Prism Version 8.3.1.

**Programming and documentation**. Statistical analyses were mainly scripted using the R statistical environment [R] version 3.6.1 along with its extension packages from the comprehensive R archive network (CRAN; https://cran.r-project.org/) and the Bioconductor project [BIOC][63]. All analyses were programmed and documented using git (https://git-scm.com) for source code management in gitlab [https://gitlab.oit.duke.edu/].

**Reporting summary**. Further information on research design is available in the Nature Research Reporting Summary linked to this article.

## Data availability

All sequencing data, including barcode sequencing, bulk RNA-sequencing, single-cell RNA-sequencing and whole-exome sequencing are available at Sequence Read Archive [https://www.ncbi.nlm.nih.gov/Traces/study/?acc=PRJNA509416]. All other data are available within the Article, Supplementary Information or available from the author upon request. Source data are provided with this paper.

## Code availability

Code for barcode mapping and data analysis is available at [https://gitlab.oit.duke.edu/dcibioinformatics/pubs/alvarez-barcode-paper].

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

## Acknowledgements

We thank Nicolas Devos from the Duke Sequencing and Genomic Technologies Core for RNA-seq library preparation and sequencing. We thank the Duke Molecular Genomics Core for single-cell RNA-seq library preparation and Vaibhav Jain for scRNA-seq data analysis. We thank Daniel Abravanel, David MacAlpine, and Bernard Mathey-Prevot for helpful discussions and critical reading of the manuscript. This work was funded by the National Cancer Institute under award numbers R01CA208042 (to J.V.A.) and F31CA220957 (to A.W.) and by startup funds from the Duke Cancer Institute, the Duke University School of Medicine and the Whitehead Foundation (to J.V.A.).

## Author contributions

J.V.A., A.W., and J.S.D. were responsible for the conception, design, and interpretation of all experiments. A.W., J.S.D., B.M., R.L., R.N., D.B.F., and N.M. performed experiments and collected data. T.D.B., H.K., and P.M. performed the barcode library preparation and sequencing. J.L., J.G, A.S., and Z.S. designed and performed the bioinformatics and statistical analysis, and K.O. supervised all bioinformatics and statistical work. A.W., J.L., K.O. and J.V.A. wrote the manuscript. J.V.A. supervised all work.

## Competing interests

The authors declare no competing interests.
