## [Peer Review File · Nature Communications]

Reviewers' comments:

Reviewer #1 (Remarks to the Author):

The study used an elegant system of ex-vivo barcoded and expanded primary tumor cells from a dox-inducible Her2 breast cancer model that were re-transplanted into non-transgenic mice. Primary tumors develop in these mice, regression of primary tumors can be induced by dox-withdrawal and after several weeks recurrence of primary tumors is observed. The study then used barcode sequencing of primary tumors, regressing tumors and recurring tumors to retrieve information on the clonal composition and thereby modes of evolution. Finally, the authors tried to identify pathways relevant for recurrence and linked to the clonal composition of recurrent tumors. The paper is clearly written, the reader has no difficulty to follow the story.

Major criticism:

1) The most important question that came to my mind was: what biology is being modeled?

Although the original model (Moody et al 2002) develops lung metastasis in 92% of mice (which were shown to be dependent on the driving oncogene Her2), metastasis and non-local residual disease are not mentioned and not addressed at all. In principle, this is a model of neo-adjuvant therapy, in which the primary tumor is systemically treated, but unlike clinical routine, surgery is subsequently NOT performed. Instead, the tumor cells are left behind until escape variants emerge. This reviewer cannot think of any clinical setting that is similar. To some extent one could argue that the residual cancer cells model systemic treatment of contralateral breast metastasis, however, I would still think that the clinical decision in this rare event would be in favor for surgery. Since the authors argue at various occasions (particularly in the discussion) that their data are relevant for metastasis, this argument is a bit in conflict with the aim of authors: they would like to understand the contribution of selection and adaptation for shaping the clonal architecture of the disease. As it is now, we do not know if the clonal architecture in the mammary gland is representative for the architecture in lung or other metastases.

I would therefore strongly recommend that the authors strengthen the link to metastasis, for example by providing information about the lung metastases that may have been found in these mice.

2) It is also a pity that the authors excluded a priori a large part of the malignant evolution of a cancer. By bar-coding fully established primary tumors instead of normal mammary cells which are then naturally selected during progression they authors lose the opportunity to truly assess the clonal evolution of a cancer. Every barcode would represent a normal cell that is being transformed subsequently and the progeny would be clearly traceable. Such an approach would much better allow identifying one bar code with one clonal cancer cell genome and identical barcodes from the primary and different metastatic sites would enable tracing the evolution of each genome. What the authors did instead is to transduce an uncharacterized primary tumor. It is not known whether this cancer was clonal, heterogeneous or extremely heterogeneous. What if the cancer was largely clonal and each bar code does not reflect the heterogeneity of the cancer at injection but just adds different "colors" to genomically identical cells? What if the cancer was genomically extremely heterogeneous at transduction and each barcode indeed marks a fully individual cell? The

interpretation of many results would be very different.

I would therefore strongly recommend that the authors add a genomic characterization of the starting population and the resulting tumors to better understand the type of evolution. For example, this would certainly help to decide whether or not Met amplifications arose independently, were acquired later or were pre-existing.

3) Some technical steps and the nomenclature should be clarified: The differentiation between regressing and recurrent tumors is not completely clear. What are the criteria for recurrent tumors? The authors call tumors at 4 and 8 weeks after dox-withdrawal regressing tumors, however, Fig. 1C shows that tumors at 8 weeks are in some mice already recurring, i.e. the relative tumor volume is already increasing again. Similarly, it was not clear to which extent the loss of clonal diversity may simply be a sensitivity problem. The authors should attempt to correlate the population size with the barcode diversity to assess whether indeed clonal selection occurred. Possibly, all clones were equally sensitive to drug-withdrawal but only the most abundant were re-detected (the tip of the iceberg). If so, conclusions about selection during withdrawal should be avoided.

4) Text and information to Fig. 2 to Fig. 4 appear a bit "verbose". While the information is interesting and the analysis is justified it could be compiled better; i.e. the information of Fig. 2 on the clonal composition of primary tumors (unique barcodes, Shannon index, dissimilarity) is contained also in Fig. 3 (clonal composition of regressing tumors) and Fig. 4 (clonal composition of recurrent tumors). As the study longitudinally tracks the clonal composition the number of unique barcodes, Shannon index, abundance and dissimilarity of primary tumors, regressing and recurrent tumors should be combined in one main figure and one supplemental figure.

5) Why were not all samples in Figure 6 and Fig. S5 analyzed for the same set of markers, i.e. all for Cdh1, Vim, Ddr2, EpCAM? Why are samples in Fig. 6 analyzed for Cdh1, EpCAM and Ddr2 and those in Fig. S5 for Cdh1 and Vim?

6) The activation of the IL6 pathway as adaptive mode for recurrence is very interesting. However, it is unclear on which data basis the IL6 pathway was selected. Were there also other differentially expressed secreted factors? If so, which and why were these excluded? This part should be studied / presented in more detail to allow readers to fully acknowledge the suggested importance.

Reviewer #2 (Remarks to the Author):

In this study conducted by Walens et al., the authors employ an inducible genetically engineered mouse (GEM) model of HER2+ breast cancer in conjunction with barcode-mediated lineage tracing to inform on how therapy (modeled here by oncogene removal) impacts clonal dynamics.

Major points:

The authors conclude from that data shown in Figs, 3C, 3D and 3F that the number of lineages and tumor heterogeneity decrease in response to therapy (oncogene removal). However, it is unclear if

this is due to loss of oncogene expression (as concluded by the authors) or if inherent differences between clones (unrelated to oncogene loss) contribute to the outcome. The importance of stabilizing clonal lineages prior to treatment has recently been demonstrated in a study by Seth et al (Seth et al., 2019) and should be performed in this study for definitive conclusions to be reached.

To better understand the impact of therapy, one needs to control for barcode changes over time, in an experiment where lineages are stabilized over time. The final aliquot of cells can then be split to form a set of similar tumors (in terms of barcodes), which will serve as controls for subsequently derived tumors. A supplementary approach would be to have a control arm (with Her expression) extending to week 4 and week 8, to control to changes over time as tumors grow. This would allow for a much more comprehensive analysis of barcode lineages and their dynamics with time and under perturbation.

The authors state in the first line of the abstract “The survival of residual tumor cells following therapy and their eventual recurrence constitutes one of the biggest obstacles to obtaining cures in breast cancer, but it remains unclear how the clonal composition of tumors changes during tumor relapse”. There are several published studies that employed barcode mediated lineage tracing to study clonal dynamics during treatment (primary, residual, recurrent), tumor engraftment as well as metastasis but only a few of these were cited in the manuscript (Bhang et al., 2015; Echeverria et al., 2019; Echeverria et al., 2018; Hata et al., 2016; Hewett et al., 2017; Lan et al., 2017; Merino et al., 2019; Nguyen et al., 2014; Nguyen et al., 2015; Nolan-Stevaux et al., 2013; Seth et al., 2019; Wagenblast et al., 2015; Wylie et al., 2017). The authors should put their findings in the context of published work and highlight any unique insights provided by their study.

Educated guesses were employed to identify the molecular basis of recurrence (i.e.: c-MET amplification) based on previous studies conducted in this model and others. It is not clear why unbiased approaches (WES) were not employed to look for other mutations or copy-number variations enriched at the time of relapse. Also, this opens an interesting question about whether C-MET amplification was selected for or was pre-existing. This can be answered by isolating barcodes shown in figure 5B from an isolate of the primary tumor as described in (Seth et al., 2019).

This study would be strengthened if preclinical trials were conducted in vivo as opposed to in vitro as reported here.

Minor points:

The authors employ a two-step PCR amplification procedure (10 cycles followed by 23 cycles) for barcode sequencing. In Fig 5B, C they use the same tumors for both barcode analysis and for monitoring c-MET amplification. It is important to include the work flow for this experiment. If qPCR was performed on DNA from the initial set, was it before or after pcr steps to enrich for barcodes? This is important because, if a major portion from unamplified DNA was set aside for qPCR and other experiments, it would lead to a decrease in the number and heterogeneity of barcodes sequenced.

Please comment on whether some tumors margins could have been left out during microdissection leading to lower representation of barcodes upon tumor shrinkage at week 4 and week 8?

Fig. 2G is hard to interpret and this data should be presented as a scatter plot.

Typo in results section line 1, para 3 (“resemble histologically tumors”)

References:

Bhang, H.E., Ruddy, D.A., Krishnamurthy Radhakrishna, V., Caushi, J.X., Zhao, R., Hims, M.M., Singh, A.P., Kao, I., Rakiec, D., Shaw, P., et al. (2015). Studying clonal dynamics in response to cancer therapy using high-complexity barcoding. *Nat Med* 21, 440-448.

Echeverria, G.V., Ge, Z., Seth, S., Zhang, X., Jeter-Jones, S., Zhou, X., Cai, S., Tu, Y., McCoy, A., Peoples, M., et al. (2019). Resistance to neoadjuvant chemotherapy in triple-negative breast cancer mediated by a reversible drug-tolerant state. *Sci Transl Med* 11.

Echeverria, G.V., Powell, E., Seth, S., Ge, Z., Carugo, A., Bristow, C., Peoples, M., Robinson, F., Qiu, H., Shao, J., et al. (2018). High-resolution clonal mapping of multi-organ metastasis in triple negative breast cancer. *Nat Commun* 9, 5079.

Hata, A.N., Niederst, M.J., Archibald, H.L., Gomez-Caraballo, M., Siddiqui, F.M., Mulvey, H.E., Maruvka, Y.E., Ji, F., Bhang, H.E., Krishnamurthy Radhakrishna, V., et al. (2016). Tumor cells can follow distinct evolutionary paths to become resistant to epidermal growth factor receptor inhibition. *Nat Med* 22, 262-269.

Hewett, D.R., Vandyke, K., Lawrence, D.M., Friend, N., Noll, J.E., Geoghegan, J.M., Croucher, P.I., and Zannettino, A.C.W. (2017). DNA Barcoding Reveals Habitual Clonal Dominance of Myeloma Plasma Cells in the Bone Marrow Microenvironment. *Neoplasia* 19, 972-981.

Lan, X., Jorg, D.J., Cavalli, F.M.G., Richards, L.M., Nguyen, L.V., Vanner, R.J., Guilhamon, P., Lee, L., Kushida, M.M., Pellacani, D., et al. (2017). Fate mapping of human glioblastoma reveals an invariant stem cell hierarchy. *Nature* 549, 227-232.

Merino, D., Weber, T.S., Serrano, A., Vaillant, F., Liu, K., Pal, B., Di Stefano, L., Schreuder, J., Lin, D., Chen, Y., et al. (2019). Barcoding reveals complex clonal behavior in patient-derived xenografts of metastatic triple negative breast cancer. *Nat Commun* 10, 766.

Nguyen, L.V., Cox, C.L., Eirew, P., Knapp, D.J., Pellacani, D., Kannan, N., Carles, A., Moksa, M., Balani, S., Shah, S., et al. (2014). DNA barcoding reveals diverse growth kinetics of human breast tumour subclones in serially passaged xenografts. *Nat Commun* 5, 5871.

Nguyen, L.V., Pellacani, D., Lefort, S., Kannan, N., Osako, T., Makarem, M., Cox, C.L., Kennedy, W., Beer, P., Carles, A., et al. (2015). Barcoding reveals complex clonal dynamics of de novo transformed

human mammary cells. *Nature* 528, 267-271.

Nolan-Stevaux, O., Tedesco, D., Ragan, S., Makhanov, M., Chenchik, A., Ruefli-Brasse, A., Quon, K., and Kassner, P.D. (2013). Measurement of Cancer Cell Growth Heterogeneity through Lentiviral Barcoding Identifies Clonal Dominance as a Characteristic of In Vivo Tumor Engraftment. *PLoS One* 8, e67316.

Seth, S., Li, C.Y., Ho, I.L., Corti, D., Loponte, S., Sapio, L., Del Poggetto, E., Yen, E.Y., Robinson, F.S., Peoples, M., et al. (2019). Pre-existing Functional Heterogeneity of Tumorigenic Compartment as the Origin of Chemoresistance in Pancreatic Tumors. *Cell Rep* 26, 1518-1532 e1519.

Wagenblast, E., Soto, M., Gutierrez-Angel, S., Hartl, C.A., Gable, A.L., Maceli, A.R., Erard, N., Williams, A.M., Kim, S.Y., Dickopf, S., et al. (2015). A model of breast cancer heterogeneity reveals vascular mimicry as a driver of metastasis. *Nature* 520, 358-362.

Wylie, A.A., Schoepfer, J., Jahnke, W., Cowan-Jacob, S.W., Loo, A., Furet, P., Marzinzik, A.L., Pelle, X., Donovan, J., Zhu, W., et al. (2017). The allosteric inhibitor ABL001 enables dual targeting of BCR-ABL1. *Nature* 543, 733-737.

Reviewer #3 (Remarks to the Author):

The authors performed cellular barcoding experiments on mouse models to assess the clonal composition of tumor and to monitor how it changes during tumor relapse (primary tumor vs residual tumor vs recurrent tumor). First, they found high similarity between clonal composition of primary tumors from 6 different individuals. Next, they show that clonal diversity decreases during consecutive phases of tumor relapse. However, in recurrent tumor samples they found unexpectedly high diversity across individuals and low concordance with primary tumors. To explain these findings, they analyzed the level of Met gene using qPCR and performed bulk gene expression analysis.

The study is well designed and the data is technically sound. The manuscript is really well written and easy to follow. The bioinformatic and statistical tools applied were mostly appropriate, but in some cases data presented are very descriptive with little statistical testing (specific comments are presented below).

My main concern is a strong individual variability that is observed in clonal composition of recurrent tumor samples. The authors nicely tried to group tumors into similar composition patterns, but in case of such strong variability I'm not sure if 12 samples are enough to provide strong evidence. Also, additional experiments done on 8 more samples don't clearly confirm the findings on initial cohort. On the other way, Echeverria et al. (*Nature Communications*, 2018) run similar barcoding experiments on PDX models of TNBC patients and they also found many variations in clonal composition between primary tumors and metastases. I encourage Authors to discuss and compare the findings.

Therefore, I invite the authors to revise their manuscript. Below are some comments and minor corrections:

- Since analyzes were based on breast cancer progression mouse model can we generalize conclusions to different breast cancer subtypes or other cancers?
- Fig 2D – horizontal line at 50% of cumulative abundance will help with the interpretation and will be consistent with findings described in the text.
- Perhaps Fig 2E and 2F could be merged into one to increase clarity. Also, Figure 2H and 2I provide the same information that Figure 2G, so Figure 2G is redundant.
- Jensen-Shannon divergence was used to check the similarity of barcode abundances between tumors. Based on that authors conclude which tumors were more similar and dissimilar, however without any statistical testing that is needed.
- Figure 5A – what is the tumor number on x-axis? What is the scale on y-axis?
- What are tumor 1668 and 1669 samples? In previous section tumors were termed from #1 to #12.
- Supplemental Figure 4A – in Figure 2A, that shows initial population of 6 tumors, we observe similar composition between tumors with the same most abundant clone (marked in green). However, in repeated experiment with 4 tumors in each case different barcode is dominant (based on color coding). Please explain.
- Figure 6A – test for significance is needed with p-values in the text.
- In Figure 6B and Supplemental Figure 5C authors provide only 2 pathways from gene set enrichment analysis. It will be nice to provide detailed results in supplement. Are the other hallmark pathways that look interesting?
- Figure 6D – is there a statistical difference in expression of IL-6 between cohorts?
- Methods/Barcode mapping: “... were mapped to the sequencing library. “. Is it a library of all 50 M unique barcodes mentioned earlier in the text?
- Methods/RNA-Seq analysis: “RNA sequences which passed quality control ...”. What was the criteria for QC? How many reads were sequenced and mapped to genome per sample?
- Methods/RNA-Seq analysis: DESeq2 was used for differential expression analysis, but I cannot find the results for all genes in the text or supplement. Were all genes analyzed or only protein-coding genes? Volcano plots comparing cohorts may be used to nicely present these results.
- In RNA-seq analysis I found 2 outcomes of interest: tumor status (primary vs recurrent) and Met-amplification status (Met-amplified or not). What was the experiment design used in DESeq2? Were there any additional covariates included?

We are grateful to the referees for their extremely thorough and thoughtful evaluation of this manuscript. In response to their helpful comments, we have performed many new experiments and have revised the text to address the issues that they have raised. We believe that the resulting revised manuscript has been substantially improved. Specific points are addressed below, and changes to the manuscript are shown in red text.

Reviewer #1

Major concerns:

- 1) *The reviewer asks that we clarify the biology that is being modeled, with a focus on explaining how our results may relate to clonal dynamics during dissemination, distant residual disease, and distant metastasis.*

We appreciate the reviewer raising this important point. We agree that the model we use in this paper is most similar to neoadjuvant therapy for primary breast cancer. We further agree that our model differs from breast cancer patients treated with neoadjuvant therapy because we do not remove the primary tumor, but instead let the tumor remain in the mammary gland and recur locally. While we agree this is an important distinction, we argue that understanding the survival and persistence of tumor cells locally is important for the following reasons:

- The response of primary breast tumors to neoadjuvant therapy is strongly associated with future risk of distant relapse, even when the residual tumor in the breast is surgically removed.
- The incidence of local recurrence is strongly associated with an increased risk of distant recurrence.

These associations suggest that the survival of cells locally in the breast in response to therapy is a surrogate for the survival of disseminated cells. In light of this, we believe that studying residual tumor survival cell and recurrence locally will provide insight into the survival and recurrence of residual cells at distant sites. We have added text to the Discussion (lines 480-493) expanding upon this important point.

We also agree that exploring the clonal dynamics of metastatic disease in this model would be useful. As the reviewer points out, tumors arising in this model frequently metastasize to the lungs, and these tumors remain dependent upon Her2 signaling (Moody et al., 2002). We have published results showing that orthotopic tumors in this model also metastasize to the lungs (Alvarez et al., 2013). While spontaneous metastasis from tumors in the mammary fat pad is the most faithful mimic of metastasis as it occurs in patients, spontaneous metastasis is an inefficient process and metastatic tumors arising spontaneously would therefore likely only contain a small number of barcodes/clones. This would preclude using a spontaneous metastasis model to study the clonal dynamics of how metastases respond to therapy.

Therefore, to study the clonal composition of metastases, and how this changes following Her2 downregulation, we used an experimental metastasis model in which cells were introduced into circulation. We injected barcoded primary tumor cells into the tail vein of mice on doxycycline. Lung metastases formed within 2-3 weeks. One cohort of mice were sacrificed with lung metastases with Her2 on (+dox), and the remaining mice had dox withdrawn from their drinking water to induce Her2 downregulation. Dox withdrawal led to Her2 downregulation as measured by luciferase imaging. While we could not assess whether Her2 downregulation led to tumor regression, mice removed from dox survived longer than mice left on dox, suggesting that Her2 downregulation led to regression of metastases (as in Moody et al.), or at least prevented the growth of metastases. This second cohort of mice was sacrificed 4 weeks following Her2 downregulation. DNA was harvested from the entire lungs in order to perform barcode sequencing. Unfortunately, despite extensive troubleshooting, we were unable to generate high-quality libraries from these samples for barcode sequencing. Therefore we cannot present data on the barcode abundance of lung metastases before and after Her2 downregulation. Nonetheless, the data we add shows that lung metastases remain dependent upon Her2 but eventually recur through Her2-independent mechanisms. These data are presented in Supplemental Figure 2.

- 2) *The reviewer points out that our interpretation of the barcoding experiments would be improved by understanding the pre-existing heterogeneity of the primary tumor. To gain insight into this, the reviewer suggests that we perform a genomic characterization of the starting primary tumor population.*

We appreciate the reviewer's point, and we agree that understanding the heterogeneity of the starting population is important. To gain insight into this, we performed two additional experiments: single-cell RNA-sequencing (scRNA-seq) and whole exome sequencing (WES) on the starting population. scRNA-seq provides, at single-cell resolution, insight into

the transcriptional heterogeneity of tumors. In contrast, WES provides data on single-nucleotide variants (SNVs) and copy-number alterations (CNAs), though the sensitivity of these techniques for detecting subclonal variants is much lower (~5% for SNVs, and ~30% for CNAs)¹ Therefore, we viewed these two approaches as complementary.

We performed scRNA-seq using the 10X Genomics platform on 5,000 tumor cells. We sequenced two primary tumor cell populations, one Met-amplified recurrent tumor cell population, and one recurrent tumor cell population without Met amplification. These results are shown in Supplemental Figure 4. This experiment showed that:

- Both primary tumor cell populations had evidence of transcriptional heterogeneity.
- Both primary tumor cell populations had subsets of cells with high expression of EMT transcripts, suggesting pre-existing heterogeneity.

Taken together, these results suggest that primary tumors have pre-existing transcriptional heterogeneity, consistent with the notion that our barcoding experiments were labeling cells with distinct phenotypic characteristics.

We also performed WES on a subset of primary and recurrent tumors from the original barcoded tumor cohort, as well as two recurrent tumor cell lines. We found that both primary and recurrent tumors had very few non-synonymous SNVs, consistent with a recent genomic characterization of Her2-driven mammary tumors². Further, we did not identify any likely driver mutations in the top 20 most frequently mutated genes in human breast cancer, with the exception of PIK3CA, which was mutated in one of the recurrent tumor cell lines (Supplemental Table 3). In contrast, we identified a number of CNAs in both primary and recurrent tumors. A limitation of this approach is that we could not determine whether these CNAs were present in all cells or in a fraction of subclones. However, taken together with the scRNA-seq described above, these results provide evidence that primary tumors have transcriptional heterogeneity as well as copy-number alterations. While we do not have evidence that any of these CNAs are subclonal, the finding that these tumors have CNAs suggests that the presence of chromosomal instability that may serve as a source of intratumor heterogeneity. Taken together, these genomic analyses suggest that DNA barcoding of primary tumor cells can label clones that are transcriptionally – and possibly genetically – distinct. The results of the genomic analysis are presented in Supplemental Figure 4 and 8 and Supplemental Table 3.

- 3) *The reviewer asks that we clarify the criteria for distinguishing residual vs. recurrent tumors in light of the fact that some tumors recur by 8 weeks. The reviewer further asks that we determine whether the apparent reduction in clonal complexity in residual tumors was due to only sequencing the most abundant clones.*

We apologize for confusion caused by the presentation of the data in Figure 1C and lack of clarity for distinguishing residual vs recurrent tumors. We have added text to Lines 73-75 describing the criteria for distinguishing residual vs. recurrent tumors. As the reviewer correctly points out, some tumors begin to recur by 8 weeks. However, these tumors were readily visible and palpable at 8 weeks and therefore would not have been considered residual tumors. We acknowledge that i) this approach biases the 8-week residual tumor cohort toward late-recurring tumors; and ii) the reduction in clonal complexity in some 8-week residual tumors could be due to incipient recurrence in a subset of clones.

We agree there is a relationship between the number of tumor cells at each stage (population size), the distribution of barcodes within a tumor (complexity), and the ability to detect rare subclones (sensitivity). In our experiments, we sequenced a fixed number of genomes from each sample (~100,000); thus, the lower limit of detection for detecting subclones is 1 in 100,000. Therefore, when we fail to detect a given barcode, that indicates that the cell marked by that barcode is present at below 1 in 100,000 cells. We have added text (Line 111) to clarify this point. To directly address the reviewer's question, we performed a simulation experiment where we randomly selected different numbers of barcodes from each sample, and measured the complexity of this subset using the Shannon index. This analysis showed that for all samples, the Shannon index held relatively constant even when considering only very low read numbers. This indicates that the reduction in clonal complexity observed in residual tumors is not simply due to detecting fewer barcodes in these samples. We have added this data to Supplementary Figure 5 and Lines 196-204 of the Results.

- 4) *The reviewer suggests that we can condense Figures 2 – 4 and make the results describing these figures more concise.*

We appreciate the reviewer making this suggestion and agree with this point. We feel that Figure 2 makes a number of important points, and also introduces the barcode analyses, and so we have elected to leave that Figure on its own. However,

we agree that Figures 3 and 4 could be combined into one figure. We have done this, and also attempted to make the results section describing these figures more concise.

5) *The reviewer points out that we used slightly different markers to assess EMT in vitro and in vivo.*

We apologize for this inconsistency. We have added new results to Supplementary Figure 10 so that all samples are analyzed for E-cadherin, Epcam, and Ddr2.

6) *The reviewer asks that we expand upon our decision to focus on IL-6 among other potentially differentially regulated secreted factors.*

Based upon the RNA-seq data from recurrent tumors, we first calculated, for every gene, the fold-change in expression between recurrent tumors without Met amplification and those with Met amplification. We then focused on secreted factors within this ranked list using a curated KEGG gene set of cytokines and cytokine receptors (KEGG_CYTOKINE_CYTOKINE_RECEPTOR_INTERACTION). The gene encoding IL6 (*Il6*) had the second-highest fold-change on this list, being expressed approximately ~15-fold higher in recurrent tumors without Met amplification than in tumors with Met amplification. (The cytokine with the highest fold-changes was Ccl17, but its receptor, Ccr4, was not expressed at all in primary or recurrent tumor cells and so we did not consider this cytokine further.) In light of this finding, and the importance of IL-6 in promoting tumor cell growth, we focused on this cytokine for future investigation. We have added an explanation of this process to the text on Lines 364-369.

Reviewer #2

Major concerns:

In this study conducted by Walens et al., the authors employ an inducible genetically engineered mouse (GEM) model of HER2+ breast cancer in conjunction with barcode-mediated lineage tracing to inform on how therapy (modeled here by oncogene removal) impacts clonal dynamics.

Major points:

- 1) *The reviewer points out the importance of ensuring that the population's clonal composition is stabilized with Her2 on, prior to conducting experiments to assess changes in clonal composition after turning Her2 off. To address this, the reviewer suggests assessing changes in barcode abundance over time in vitro, and also measuring barcode abundance in tumors in vivo grown for an additional 4-8 weeks.*

We appreciate the reviewer raising this important point, and we completely agree with these suggestions. First, we note that the injected cell population we used as a reference in the initial submitted manuscript was grown for 12 population doublings between puromycin selection and sequencing (at which point these cells were also frozen). This provides some amount of time for clonal stabilization. To directly respond to the reviewer's request, we thawed these cells, and passaged them for an additional 10 population doublings, and then repeated the barcode sequencing. These results, which are shown in Supplemental Figure 1, demonstrate that there was minimal clonal drift during passaging. We still detected approximately 13,000 barcodes, and the Shannon index was 8. In addition, the dissimilarity index revealed that this samples had highly correlated barcode distributions to the original sequenced sample.

We also performed the second suggested experiment. We grew primary tumors until one dimension reached the maximum tumor diameter permitted under our IACUC protocol (15 mm). On average, these tumors were 9-fold larger in volume than primary tumors described in our initial submitted manuscript, which were ~5 mm in diameter. To distinguish this cohort, we refer to these tumors as "late primary tumors." As the reviewer suggests, these late primary tumors had fewer unique barcodes and a lower Shannon index compared to primary tumors. However, the clonal composition of these tumors was quite distinct from that of 4- and 8-week residual tumors. Based upon this, we conclude that both continued tumor growth in the presence of Her2, as well as tumor regression induced by Her2 downregulation, are accompanied by decreased clonal complexity. However, these different scenarios impose different selective pressures and therefore select for different clones. These results are shown in Supplemental Figure 6.

- 2) *The reviewer cites a number of studies that used barcoding to study tumor formation and metastasis that we failed to cite, and asks that we put our findings in the context of these other studies.*

We apologize for failing to cite these important studies. First, we have cited all of these papers in the Results section where we introduce previous work on barcoding. Second, we cite many of these papers at several points in the Discussion as we discuss our work and put it into context.

- 3) *Educated guesses were employed to identify the molecular basis of recurrence (i.e.: c-MET amplification) based on previous studies conducted in this model and others. It is not clear why unbiased approaches (WES) were not employed to look for other mutations or copy-number variations enriched at the time of relapse.*

As described in response to Reviewer #1's Point #2, we have performed WES on a subset of primary and recurrent tumors from the original cohort, as well as two recurrent tumor cell lines. We found that both primary and recurrent tumors had very few non-synonymous SNVs, consistent with a recent genomic characterization of Her2-driven mammary tumors². Further, we did not identify any likely driver mutations in the top 20 most frequently mutated genes in human breast cancer, with the exception of PIK3CA, which was mutated in one of the recurrent tumor cell lines. In contrast, we identified a number of CNAs in both Met-amplified and non-amplified recurrent tumors. While we could not identify any candidate driver oncogenes with focal, high-level amplification in recurrent tumors that may drive recurrence (besides Met itself), this analysis does indicate that recurrent tumors have more CNAs as compared to primary tumors. The results of the genomic analysis are presented in Supplemental Figure 8 and Supplemental Table 3.

- 4) *Also, this opens an interesting question about whether C-MET amplification was selected for or was pre-existing. This can be answered by isolating barcodes shown in figure 5B from an isolate of the primary tumor as described*

in (Seth et al., 2019).

We agree that the elegant approach devised by Seth et al. is, to our knowledge, the only way to conclusively demonstrate whether Met amplification arises from pre-existing Met-amplified cells. Unfortunately, we did not have the resources or technical expertise to perform the prospective isolation of individual barcoded clones as described in Seth et al, which involves novel library prep methods and extensive next-generation sequencing. Nonetheless, we agree that this would be a definitive experiment and we have added text to the Discussion (Lines 400-405) addressing this point.

- 5) *This study would be strengthened if preclinical trials were conducted in vivo as opposed to in vitro as reported here.*

We appreciate the reviewer's suggestion. We added an experiment demonstrating that crizotinib treatment inhibited the growth of Met-amplified recurrent tumor cells in vivo (Supplemental Figure 8). We have also added in vivo evidence that recurrent tumors undergo EMT in vivo (Supplemental Figure 10).

Minor points:

- 1) *The authors employ a two-step PCR amplification procedure (10 cycles followed by 23 cycles) for barcode sequencing. In Fig 5B, C they use the same tumors for both barcode analysis and for monitoring c-MET amplification. It is important to include the work flow for this experiment. If qPCR was performed on DNA from the initial set, was it before or after pcr steps to enrich for barcodes? This is important because, if a major portion from unamplified DNA was set aside for qPCR and other experiments, it would lead to a decrease in the number and heterogeneity of barcodes sequenced.*

We always performed PCR on 640 ng of genomic DNA (~100,000 genomes) for sequencing. We obtained more than this amount of DNA from all samples, including residual tumors: all residual tumor samples yielded >10 ug DNA, or ~1.5 million genomes, and primary and recurrent tumors yielded much more DNA. We used qPCR with a probe specific for the barcode cassette to quantify tumor cells, and found that all residual tumors had greater than 1 million tumor cells. Thus, the qPCR and other genomic analyses (WES) did not limit the amount of DNA that went into the barcode PCR reactions. We have added text to the Methods section (Lines 544-546) to clarify this.

- 2) *Please comment on whether some tumors margins could have been left out during microdissection leading to lower representation of barcodes upon tumor shrinkage at week 4 and week 8?*

When we dissected residual tumors, we included ample margins around the fluorescent tumor to ensure that we did not miss any tumor cells and therefore barcodes. We have added text to the Methods section (Lines 529-530) to clarify this.

- 3) *Fig. 2G is hard to interpret and this data should be presented as a scatter plot.*

We have changed this so that the y-axis is shown on a log scale, which we believe makes this data easier to see.

- 4) *Typo in results section line 1, para 3 (“resemble histologically tumors”)*

We have fixed this error.

Reviewer #3

The authors performed cellular barcoding experiments on mouse models to assess the clonal composition of tumor and to monitor how it changes during tumor relapse (primary tumor vs residual tumor vs recurrent tumor). First, they found high similarity between clonal composition of primary tumors from 6 different individuals. Next, they show that clonal diversity decreases during consecutive phases of tumor relapse. However, in recurrent tumor samples they found unexpectedly high diversity across individuals and low concordance with primary tumors. To explain these findings, they analyzed the level of Met gene using qPCR and performed bulk gene expression analysis.

The study is well designed and the data is technically sound. The manuscript is really well written and easy to follow. The bioinformatic and statistical tools applied were mostly appropriate, but in some cases data presented are very descriptive with little statistical testing (specific comments are presented below).

My main concern is a strong individual variability that is observed in clonal composition of recurrent tumor samples. The authors nicely tried to group tumors into similar composition patterns, but in case of such strong variability I'm not sure if 12 samples are enough to provide strong evidence. Also, additional experiments done on 8 more samples don't clearly confirm the findings on initial cohort. On the other way, Echeverria et al. (Nature Communications, 2018) run similar barcoding experiments on PDX models of TNBC patients and they also found many variations in clonal composition between primary tumors and metastases. I encourage Authors to discuss and compare the findings.

Therefore, I invite the authors to revise their manuscript. Below are some comments and minor corrections:

- 1) *The reviewer notes that there is strong individual variability in clonal composition of recurrent tumors, similar to recent observations in TNBC, and asks that we discuss our findings in the context of this work.*

We appreciate the reviewer raising this point. We have added a discussion of this topic to Lines 423-492 of the Discussion:

One striking finding from these studies is the apparent diversity of recurrence paths accessible to tumors. While Met-amplified recurrent tumors had similar gene expression patterns to one another, recurrent tumors lacking gene Met amplification were quite heterogeneous with respect to both clonal composition and gene expression profiles. These results are similar to studies analyzing clonal architecture in TNBC metastasis, which observed substantial variations in the clonal complexity of metastases as compared to corresponding primary tumors. Future work will be needed to define the common phenotypic properties shared by tumors with distinct clonal architecture but similar propensities to recur.

- 2) *Since analyzes were based on breast cancer progression mouse model can we generalize conclusions to different breast cancer subtypes or other cancers?*

We appreciate the reviewer raising this important point. We have added text to the Discussion (paragraph beginning on line 423) speculating on whether these conclusions are generalizable and putting them in the context of other studies on different cancer types.

- 3) *Fig 2D – horizontal line at 50% of cumulative abundance will help with the interpretation and will be consistent with findings described in the text.*

We appreciate this suggestion – we have added a horizontal line to this Figure.

- 4) *Perhaps Fig 2E and 2F could be merged into one to increase clarity. Also, Figure 2H and 2I provide the same information that Figure 2G, so Figure 2G is redundant.*

In response to this comment and Reviewer #2's similar comment, we have changed the y-axis for Figure 2E, 2F, and 2G to be in log scale. We think this improves the clarity of these panels. While we agree that Figure 2G provides similar information to 2E-I, we feel this way of displaying the data nicely shows the difference in barcode abundance between tumors and injected cells.

- 5) *Jensen-Shannon divergence was used to check the similarity of barcode abundances between tumors. Based on that authors conclude which tumors were more similar and dissimilar, however without any statistical testing that is needed.*

We apologize for the omission. We have added statistical testing of the differences in Jensen-Shannon divergence between cohorts (Lines 147 and 209-210).

6) *Figure 5A – what is the tumor number on x-axis? What is the scale on y-axis?*

We apologize for the omissions. We have added tumor numbers to the x-axis. We changed the scale of the y-axis so that it displays the absolute number of copies of the *Met* gene (as opposed to relative *Met* amplification, as originally presented).

7) *What are tumor 1668 and 1669 samples? In previous section tumors were termed from #1 to #12.*

We apologize for the oversight. 1668 and 1669 are additional barcoded recurrent tumors that were generated in order to perform functional in vitro studies. We could not perform these experiments on Recurrent Tumors #1-12, since the entire tumor was used to isolate gDNA. We have added Supplementary Table 8 that has information on all the tumors and tumor cell lines used in this paper and their corresponding figures.

8) *Supplemental Figure 4A – in Figure 2A, that shows initial population of 6 tumors, we observe similar composition between tumors with the same most abundant clone (marked in green). However, in repeated experiment with 4 tumors in each case different barcode is dominant (based on color coding). Please explain.*

We appreciate the reviewer raising this point. Supplemental Figure 4A shows results from a second donor tumor, primary #2. (See Supplementary Table 8 for information on each cell line.) It is possible that, unlike primary tumor cell line #1, this second primary tumor cell line does not yield primary orthotopic tumors with consistent clonal composition. Alternatively, it may be necessary to sequence more than 4 tumors to see a reproducible pattern.

9) *Figure 6A – test for significance is needed with p-values in the text.*

We apologize for the omission. We have added p-values to the graphs and specified the statistical test used in the figure legend.

10) *In Figure 6B and Supplemental Figure 5C authors provide only 2 pathways from gene set enrichment analysis. It will be nice to provide detailed results in supplement. Are the other hallmark pathways that look interesting?*

We have added Supplementary Tables 4 and 5 with GSEA results.

11) *Figure 6D – is there a statistical difference in expression of IL-6 between cohorts?*

The difference in IL-6 expression is significant between cohorts. We have added the p-value to the graph and in the text (Line 369).

12) *Methods/Barcode mapping: “... were mapped to the sequencing library. “ Is it a library of all 50 M unique barcodes mentioned earlier in the text?*

Yes, the sequencing reads were mapped to the entire library of 50M unique barcodes that are present in this library. We have clarified this in the Methods.

13) *Methods/RNA-Seq analysis: “RNA sequences which passed quality control ...”. What was the criteria for QC? How many reads were sequenced and mapped to genome per sample?*

We have changed the text to include QC criteria (Phred score>10) and include a figure (Supplemental Figure 11) with the alignment and QC summary.

14) *Methods/RNA-Seq analysis: DESeq2 was used for differential expression analysis, but I cannot find the results for all genes in the text or supplement. Were all genes analyzed or only protein-coding genes? Volcano plots comparing cohorts may be used to nicely present these results.*

We have included two files with the differential gene expression analysis (as described below).

15) *In RNA-seq analysis I found 2 outcomes of interest: tumor status (primary vs recurrent) and Met-amplification status (Met-amplified or not). What was the experiment design used in DESeq2? Were there any additional covariates included?*

We appreciate the reviewer making this important point. We performed DESeq2 both comparing primary vs. recurrent tumors and Met-amplified recurrent tumors vs. non-amplified recurrent tumors. As described above, these files are uploaded as Supplementary Tables 6 and 7.

References

- 1 Jiang Y, Wang, R, Urrutia, E, Anastopoulos, IN, Nathanson, KL & Zhang, NR. CODEX2: full-spectrum copy number variation detection by high-throughput DNA sequencing. *Genome Biol* **19**, 202, doi:10.1186/s13059-018-1578-y (2018).
- 2 Rennhack JP, To, B, Swiatnicki, M, Dulak, C, Ogrodzinski, MP, Zhang, Y, Li, C, Bylett, E, Ross, C, Szczepanek, K, Hanrahan, W, Jayatissa, M, Lunt, SY, Hunter, K & Andrechek, ER. Integrated analyses of murine breast cancer models reveal critical parallels with human disease. *Nat Commun* **10**, 3261, doi:10.1038/s41467-019-11236-3 (2019).

Reviewers' comments:

Reviewer #1 (Remarks to the Author):

My criticism in the first round was the following:

1) The most important question that came to my mind was: what biology is being modeled? Although the original model (Moody et al 2002) develops lung metastasis in 92% of mice (which were shown to be dependent on the driving oncogene Her2), metastasis and non-local residual disease are not mentioned and not addressed at all. In principle, this is a model of neo-adjuvant therapy, in which the primary tumor is systemically treated, but unlike clinical routine, surgery is subsequently NOT performed. Instead, the tumor cells are left behind until escape variants emerge. I cannot think of any clinical setting that is similar. To some extent one could argue that the residual cancer cells model systemic treatment of contralateral breast metastasis, however, I would still think that the clinical decision in this rare event would be in favor for surgery. Since the authors argue at various occasions (particularly in the discussion) that their data are relevant for metastasis, this argument is a bit in conflict with the aim of authors: they would like to understand the contribution of selection vs. adaptation (which would occur in response to conditions at the metastatic site!) for shaping the clonal architecture of the disease. As it is now, we do not know if the clonal architecture in the mammary gland is representative for the architecture in lung or other metastases. I would therefore strongly recommend that the authors strengthen the link to metastasis, for example by providing information about the lung metastases that may have been found in these mice.

The authors' reply to this point is the following:

"We appreciate the reviewer raising this important point. We agree that the model we use in this paper is most similar to neoadjuvant therapy for primary breast cancer. We further agree that our model differs from breast cancer patients treated with neoadjuvant therapy because we do not remove the primary tumor, but instead let the tumor remain in the mammary gland and recur locally. While we agree this is an important distinction, we argue that understanding the survival and persistence of tumor cells locally is important for the following reasons:

- The response of primary breast tumors to neoadjuvant therapy is strongly associated with future risk of distant relapse, even when the residual tumor in the breast is surgically removed.
- The incidence of local recurrence is strongly associated with an increased risk of distant recurrence.

These associations suggest that the survival of cells locally in the breast in response to therapy is a surrogate for the survival of disseminated cells. In light of this, we believe that studying residual tumor survival cell and recurrence locally will provide insight into the survival and recurrence of residual cells at distant sites. We have added text to the Discussion (lines 480-493) expanding upon this important point. We also agree that exploring the clonal dynamics of metastatic disease in this model would be useful. As the reviewer points out, tumors arising in this model frequently metastasize to the lungs, and these tumors remain dependent upon Her2 signaling (Moody et al., 2002). We have published results showing that orthotopic tumors in this model also metastasize to the lungs (Alvarez et al., 2013). While spontaneous metastasis from tumors in the mammary fat pad is the most faithful mimic of metastasis as it occurs in patients, spontaneous metastasis is an

inefficient process and metastatic tumors arising spontaneously would therefore likely only contain a small number of barcodes/clones. This would preclude using a spontaneous metastasis model to study the clonal dynamics of how metastases respond to therapy."

This answer is confusing. I understood that the authors intend to use the barcoding method in order to better study the clonal dynamics than it is possible in patients. However, now they state that a model mimicking the patient situation would not be useful to study the clonal dynamics. Instead, they use experimental metastasis via tail vein injection, however, this experiment failed to generate any answer due to technical reasons. I feel sorry for the authors, however, I am still left with the question what I have learned from the study.

I would have preferred an experiment that is as close to the patient situation as possible and study clonal evolution. The metastatic inefficiency of spontaneous metastasis that the authors see as disadvantage is part of the clonal selection which their barcoding approach could have uncovered. Therapy of patients has two aspects: selection pressure by reducing population size via surgery and selection pressure via systemic treatment (here by dox withdrawal). Selection via surgery acts on cells that need to survive in a hostile environment and this process may have fundamental consequences for the selection via systemic treatment. (Imaging that the first selection confers differing fitness (dis)advantage for the second selection step!). Therefore, I think the premise of the authors ('clonal dynamics at the primary site' models 'clonal dynamics at metastatic sites') needs to be proven. Their technology might have the power to do so, but it is not being used.

The second point, I asked the authors to address was the following:

2) It is also a pity that the authors excluded a priori a large part of the malignant evolution of a cancer. By bar-coding fully established primary tumors instead of normal mammary cells which are then naturally selected during progression they authors lose the opportunity to truly assess the clonal evolution of a cancer. Every barcode would represent a normal cell that is being transformed subsequently and the progeny would be clearly traceable. Such an approach would much better allow identifying one bar code with one clonal cancer cell genome and identical barcodes from the primary and different metastatic sites would enable tracing the evolution of each genome. What the authors did instead is to transduce an uncharacterized primary tumor. It is not known whether this cancer was clonal, heterogeneous or extremely heterogeneous. What if the cancer was largely clonal and each bar code does not reflect the heterogeneity of the cancer at injection but just adds different "colors" to genomically identical cells? What if the cancer was genomically extremely heterogeneous at transduction and each barcode indeed marks a fully individual cell? The interpretation of many results would be very different.

I would therefore strongly recommend that the authors add a genomic characterization of the starting population and the resulting tumors to better understand the type of evolution. Ideally, these would be single cell analyses (sc-sequencing or FISH or other methods) to get an idea about the heterogeneity. For example, this would certainly help to decide whether or not Met amplifications arose independently, were acquired later or were pre-existing.

The authors reply:

"We appreciate the reviewer's point, and we agree that understanding the heterogeneity of the starting population is important. To gain insight into this, we performed two additional experiments:

single-cell RNA-sequencing (scRNA-seq) and whole exome sequencing (WES) on the starting population. scRNA-seq provides, at single-cell resolution, insight into the transcriptional heterogeneity of tumors. In contrast, WES provides data on single-nucleotide variants (SNVs) and copynumber alterations (CNAs), though the sensitivity of these techniques for detecting subclonal variants is much lower (~5% for SNVs, and ~30% for CNAs)¹ Therefore, we viewed these two approaches as complementary. We performed scRNA-seq using the 10X Genomics platform on 5,000 tumor cells. We sequenced two primary tumor cell populations, one Met-amplified recurrent tumor cell population, and one recurrent tumor cell population without Met amplification. These results are shown in Supplemental Figure 4. This experiment showed that:

- Both primary tumor cell populations had evidence of transcriptional heterogeneity.
- Both primary tumor cell populations had subsets of cells with high expression of EMT transcripts, suggesting preexisting heterogeneity. Taken together, these results suggest that primary tumors have pre-existing transcriptional heterogeneity, consistent with the notion that our barcoding experiments were labeling cells with distinct phenotypic characteristics."

While I appreciate that the authors performed an expensive single cell experiment, I wonder what they expected to find. Why did they perform RNA-seq? My question was: are the bar-codes linked to differing genomes, i.e. do they represent different clones? How could an RNA-seq experiment answer that question?

In the current manuscript the barcodes are equated with clones, however, there is no data set provided that would support this assumption. At minimum, it should be made clear throughout the text that the number of barcodes does not inform about the number of clones, i.e. the terms "clones", "clonal complexity", "clonal diversity" etc... should be deleted. The text should be rephrased and instead the authors should write " barcodes", "barcode complexity", "barcode diversity" etc., if the data is based on the analysis of barcodes. The sc-RNA analysis informs to some extent about the transcriptionally heterogeneity of explanted tumors that were cultured. However, as this analysis was not combined with information on the genotypes or barcodes of the cells, it is still unclear whether the transcriptional heterogeneity reflects the selection of different clones or rather adaptation/plasticity of the same clone. Therefore, I have difficulties to follow the argument of the authors that the barcoding experiments were labeling cells with distinct phenotypic characteristics. I cannot see in supplementary figure 4 that different barcodes are linked with the different transcriptional phenotypes.

In summary, my two major points were not addressed. In both cases experiments were performed that could not answer my questions. Therefore, many claims are not sufficiently supported in my opinion and it remains unclear what we have learned from the paper.

Reviewer #2 (Remarks to the Author):

The revised manuscript is improved by the additional experiments and text revisions.

Reviewer #3 (Remarks to the Author):

I thank the authors for satisfactorily answering all my questions and doubts. In the revised version, the authors have significantly expanded the manuscript with new data and analyzes, and added results of statistical testing where needed. In my opinion the manuscript is now acceptable for publication after applying these minor corrections:

- Line 309 – text describes results shown on Supplementary Figure 8C, not 8B
- Line 606 – what specific version of Cell Ranger was used? It is important, since different versions may use different genome and transcriptome annotation
- Line 616 – it is written that cells are visualized using t-SNE, however Supplementary Figure 4 shows UMAP plots, not t-SNE.

Reviewer #1 (Remarks to the Author):

“My criticism in the first round was the following:

1) The most important question that came to my mind was: what biology is being modeled? Although the original model (Moody et al 2002) develops lung metastasis in 92% of mice (which were shown to be dependent on the driving oncogene Her2), metastasis and non-local residual disease are not mentioned and not addressed at all. In principle, this is a model of neo-adjuvant therapy, in which the primary tumor is systemically treated, but unlike clinical routine, surgery is subsequently NOT performed. Instead, the tumor cells are left behind until escape variants emerge. I cannot think of any clinical setting that is similar. To some extent one could argue that the residual cancer cells model systemic treatment of contralateral breast metastasis, however, I would still think that the clinical decision in this rare event would be in favor for surgery. Since the authors argue at various occasions (particularly in the discussion) that their data are relevant for metastasis, this argument is a bit in conflict with the aim of authors: they would like to understand the contribution of selection vs. adaptation (which would occur in response to conditions at the metastatic site!) for shaping the clonal architecture of the disease. As it is now, we do not know if the clonal architecture in the mammary gland is representative for the architecture in lung or other metastases.

I would therefore strongly recommend that the authors strengthen the link to metastasis, for example by providing information about the lung metastases that may have been found in these mice.

The authors' reply to this point is the following:

"We appreciate the reviewer raising this important point. We agree that the model we use in this paper is most similar to neoadjuvant therapy for primary breast cancer. We further agree that our model differs from breast cancer patients treated with neoadjuvant therapy because we do not remove the primary tumor, but instead let the tumor remain in the mammary gland and recur locally. While we agree this is an important distinction, we argue that understanding the survival and persistence of tumor cells locally is important for the following reasons:

- The response of primary breast tumors to neoadjuvant therapy is strongly associated with future risk of distant relapse, even when the residual tumor in the breast is surgically removed.*

- The incidence of local recurrence is strongly associated with an increased risk of distant recurrence.*

These associations suggest that the survival of cells locally in the breast in response to therapy is a surrogate for the survival of disseminated cells. In light of this, we believe that studying residual tumor survival cell and recurrence locally will provide insight into the survival and recurrence of residual cells at distant sites. We have added text to the Discussion (lines 480-493) expanding upon this important point. We also agree that exploring the clonal dynamics of metastatic disease in this model would be useful. As the reviewer points out, tumors arising in this model frequently metastasize to the lungs, and these tumors remain dependent upon Her2 signaling (Moody et al., 2002). We have published results showing that orthotopic tumors in this model also metastasize to the lungs (Alvarez et al., 2013). While spontaneous metastasis from tumors in the mammary fat pad is the most faithful mimic of metastasis as it occurs in patients, spontaneous metastasis is an inefficient process and metastatic tumors arising spontaneously would therefore likely only contain a small number of barcodes/clones. This would preclude using a spontaneous metastasis model to study the clonal dynamics of how metastases respond to therapy."

This answer is confusing. I understood that the authors intend to use the barcoding method in order to better study the clonal dynamics than it is possible in patients. However, now they state that a model mimicking the patient situation would not be useful to study the clonal dynamics. Instead, they use experimental metastasis via tail vein injection, however, this experiment failed to generate any answer due to technical reasons. I feel sorry for the authors, however, I am still left with the question what I have learned from the study.

I would have preferred an experiment that is as close to the patient situation as possible and study clonal evolution. The metastatic inefficiency of spontaneous metastasis that the authors see as disadvantage is part of the clonal selection which their barcoding approach could have uncovered. Therapy of patients has two aspects: selection pressure by reducing population size via surgery and selection pressure via systemic

treatment (here by dox withdrawal). Selection via surgery acts on cells that need to survive in a hostile environment and this process may have fundamental consequences for the selection via systemic treatment. (Imaging that the first selection confers differing fitness (dis)advantage for the second selection step!). Therefore, I think the premise of the authors ('clonal dynamics at the primary site' models 'clonal dynamics at metastatic sites') needs to be proven. Their technology might have the power to do so, but it is not being used."

Response: There are both conceptual and technical reasons why we disagree with the reviewer's comment.

Conceptual: While we agree with the reviewer that understanding how disseminated tumor cells respond to therapy is of critical importance in breast cancer, it does not follow that we cannot therefore learn anything from studying a tumor's response to therapy at the primary site. This argument proves too much: it would mean that every preclinical study examining response to therapy at the primary site is not relevant. Further, it would render meaningless the entire clinical area of neoadjuvant therapy, which examines the response of breast tumors to chemotherapy in the breast, prior to surgery.

In actuality, the field has learned a lot from studying how tumors respond to therapy and develop resistance at the primary site. There are countless examples of publications that prove this point, but here I pick two studies among many potential examples: Relevant to the current study, Echeverria et al. used DNA barcoding to study the response of triple-negative breast cancers to chemotherapy *in the mammary gland* (Echeverria et al., 2019). And in a clinical study, Balko et al. analyzed residual tumor cells that remain following neoadjuvant therapy *in the breast* and identified activation of targetable pathways (Balko et al., 2014). Indeed, as we describe in the Discussion (lines 483-495), the observation that response to neoadjuvant therapy in the breast predicts the likelihood of distant recurrence proves the point that local and distant response to therapy are linked. Thus, we believe that the literature strongly suggests that studying response to therapy in the primary site offers relevant insight into therapy resistance. Nonetheless, we have added a sentence to the last line of the Discussion emphasizing the reviewer's point on importance of studying the response of disseminated tumor cells to therapy.

Technical: Barcoding can be used to study the clonal dynamics of metastatic dissemination and seeding (Echeverria et al., 2018; Wagenblast et al., 2015). Barcoding can also be used to study the clonal dynamics of a tumor's response to therapy (Bhang et al., 2015; Hata et al., 2016; Seth et al., 2019; Wylie et al., 2017), which is the focus of our manuscript. However, given current technology, barcoding cannot be used for both: that is, it is not possible to use barcoding to study the clonal dynamics of how disseminated tumor cells respond to therapy. This is because the large bottleneck imposed by dissemination in experimental models precludes having a diversity of barcodes in metastases, which is an essential precondition for studying clonal dynamics of these metastases' response to therapy. (To overcome this, it would be necessary to develop an inducible barcode system, in which random barcodes could be generated in metastases following dissemination in vivo. These approaches are only now being developed, and we believe this is beyond the scope of the current manuscript.) Thus, it is not currently possible to perform the experiment that would address the reviewer's concern.

The second point, I asked the authors to address was the following:

2) It is also a pity that the authors excluded a priori a large part of the malignant evolution of a cancer. By barcoding fully established primary tumors instead of normal mammary cells which are then naturally selected during progression they authors lose the opportunity to truly assess the clonal evolution of a cancer. Every barcode would represent a normal cell that is being transformed subsequently and the progeny would be clearly traceable. Such an approach would much better allow identifying one bar code with one clonal cancer cell genome and identical barcodes from the primary and different metastatic sites would enable tracing the evolution of each genome.

What the authors did instead is to transduce an uncharacterized primary tumor. It is not known whether this cancer was clonal, heterogeneous or extremely heterogeneous. What if the cancer was largely clonal and each bar code does not reflect the heterogeneity of the cancer at injection but just adds different "colors" to

genomically identical cells? What if the cancer was genomically extremely heterogeneous at transduction and each barcode indeed marks a fully individual cell? The interpretation of many results would be very different. I would therefore strongly recommend that the authors add a genomic characterization of the starting population and the resulting tumors to better understand the type of evolution. Ideally, these would be single cell analyses (sc-sequencing or FISH or other methods) to get an idea about the heterogeneity. For example, this would certainly help to decide whether or not Met amplifications arose independently, were acquired later or were pre-existing.

The authors reply:

"We appreciate the reviewer's point, and we agree that understanding the heterogeneity of the starting population is important. To gain insight into this, we performed two additional experiments: single-cell RNA-sequencing (scRNA-seq) and whole exome sequencing (WES) on the starting population. scRNA-seq provides, at single-cell resolution, insight into the transcriptional heterogeneity of tumors. In contrast, WES provides data on single-nucleotide variants (SNVs) and copynumber alterations (CNAs), though the sensitivity of these techniques for detecting subclonal variants is much lower (~5% for SNVs, and ~30% for CNAs). Therefore, we viewed these two approaches as complementary. We performed scRNA-seq using the 10X Genomics platform on 5,000 tumor cells. We sequenced two primary tumor cell populations, one Met-amplified recurrent tumor cell population, and one recurrent tumor cell population without Met amplification. These results are shown in Supplemental Figure 4. This experiment showed that:

- Both primary tumor cell populations had evidence of transcriptional heterogeneity.*
- Both primary tumor cell populations had subsets of cells with high expression of EMT transcripts, suggesting preexisting heterogeneity. Taken together, these results suggest that primary tumors have pre-existing transcriptional heterogeneity, consistent with the notion that our barcoding experiments were labeling cells with distinct phenotypic characteristics."*

While I appreciate that the authors performed an expensive single cell experiment, I wonder what they expected to find. Why did they perform RNA-seq? My question was: are the bar-codes linked to differing genomes, i.e. do they represent different clones? How could an RNA-seq experiment answer that question?

In the current manuscript the barcodes are equated with clones, however, there is no data set provided that would support this assumption. At minimum, it should be made clear throughout the text that the number of barcodes does not inform about the number of clones, i.e. the terms "clones", "clonal complexity", "clonal diversity" etc... should be deleted. The text should be rephrased and instead the authors should write "barcodes", "barcode complexity", "barcode diversity" etc., if the data is based on the analysis of barcodes. The sc-RNA analysis informs to some extent about the transcriptionally heterogeneity of explanted tumors that were cultured. However, as this analysis was not combined with information on the genotypes or barcodes of the cells, it is still unclear whether the transcriptional heterogeneity reflects the selection of different clones or rather adaptation/plasticity of the same clone. Therefore, I have difficulties to follow the argument of the authors that the barcoding experiments were labeling cells with distinct phenotypic characteristics. I cannot see in supplementary figure 4 that different barcodes are linked with the different transcriptional phenotypes.

In summary, my two major points were not addressed. In both cases experiments were performed that could not answer my questions. Therefore, many claims are not sufficiently supported in my opinion and it remains unclear what we have learned from the paper.

Response: There is both a semantic and substantive issue raised by the reviewer's concern.

Semantic issue: The reviewer suggests that the term "clone" should only be used to refer to genetically distinct cells. We disagree. It is common in the field to refer to all cells marked by a single barcode as a "clone" even in the absence of genomic characterization of these cells. Examples of this usage include Seth et al., Echeverria et al. 2018 and 2019, Bhang et al., Nguyen et al., and Wylie et al., among others (Bhang et al., 2015; Echeverria et

al., 2019; Echeverria et al., 2018; Nguyen et al., 2014; Seth et al., 2019; Wylie et al., 2017). Indeed, one of the first studies to use barcoding, Bhang et al., named their barcode library “ClonTracer” (Bhang et al., 2015). In this context, “clone” is used to denote common ancestry among cells with given barcode, NOT to argue that cells with different barcodes are genetically distinct. (We recognize that in studies using next-generation sequencing of human tumors, “clone” refers to genetically distinct cancer cells. However, this is because the only way to distinguish clones from one another using sequencing data is through the presence of mutations that are present in one clone but not another.) Thus, we believe that using “clone” to refer to a population of cells sharing the same barcode is accepted practice in the field. However, we have added a sentence to the Introduction (Lines 49-51) clarifying this use of the term.

Substantive issue: We agree that the substantive point raised by the reviewer – “Does each barcode mark a genetically distinct cell?” – is an interesting question. However, it is not technically feasible to answer this question, since it would require performing single-cell whole-genome sequencing on thousands of cells in parallel, which is not currently possible. More importantly, the majority of our significant findings would not be affected by knowing whether each barcode marks a genetically distinct cell. In fact, many of our novel findings center around adaptation – that is, how nongenetic events contribute to recurrence. Specifically, we believe our study makes five significant findings:

1. Tumor dormancy was accompanied by a reduction in clonal complexity and a continued attrition of a subset of clones. This suggests that, rather than being a static stage, tumor dormancy is a dynamic process shaped by ongoing selective pressures during residual disease.
2. The regrowth of dormant cells into recurrent tumors followed several distinct paths, yielding recurrent tumors with very different clonal architecture. One subset of recurrent tumors – clonal tumors – was composed of one or two dominant clones. A second group of tumors – polyclonal tumors – had thousands of equally distributed clones.
3. Clonal recurrent tumors were driven by selection for Met-amplified clones. Each Met-amplified recurrent tumor was composed of a different dominant clone(s), suggesting that Met amplification occurs *de novo*.
4. Polyclonal recurrent tumors were driven by an adaptive, tumor-wide reactivation of dormant tumor cells, yielding recurrent tumors whose clonal makeup closely resembled primary tumors. These polyclonal recurrent tumors represent a novel mode of tumor relapse, and we identified an autocrine acting IL-6-Jak-Stat3 pathway as a likely driver of these tumors.
5. Clonal recurrent tumors and polyclonal recurrent tumors had distinct genetic alterations, different gene expression patterns, and different drug sensitivities.

Of these five significant findings, only Point #3 would benefit from knowing whether each barcode marks a genetically distinct cell.

Finally, we note that a more realistic experiment to get at this question was raised by Reviewer #2, who suggested using an approach developed by Seth et al. to prospectively isolate specific barcodes (e.g., barcodes marking Met-amplified clones) and perform genomic characterization of these clones. This would more directly assess whether Met amplification occurs *de novo* or arise from pre-existing clones. However, even this approach would only characterize a small number of subclones (for instance, Seth et al. sequenced 12 isolated barcode lineages), and therefore would still not satisfy Reviewer #1’s concern.

In summary, we feel that the reviewer’s second concern does not limit the significance of our manuscript and is not technically feasible to address.

Reviewer #2 (Remarks to the Author):

The revised manuscript is improved by the additional experiments and text revisions.

Reviewer #3 (Remarks to the Author):

I thank the authors for satisfactorily answering all my questions and doubts. In the revised version, the authors have significantly expanded the manuscript with new data and analyzes, and added results of statistical testing where needed. In my opinion the manuscript is now acceptable for publication after applying these minor corrections:

- Line 309 – text describes results shown on Supplementary Figure 8C, not 8B

Response: We have corrected this typo.

- Line 606 – what specific version of Cell Ranger was used? It is important, since different versions may use different genome and transcriptome annotation

Response: We have added the version number of Cell Ranger.

- Line 616 – it is written that cells are visualized using t-SNE, however Supplementary Figure 4 shows UMAP plots, not t-SNE.

Response: We apologize for this error. We have corrected the Methods to clarify that UMAP was used for visualization.

References

Balko JM, Giltneane JM, Wang K, Schwarz LJ, Young CD, Cook RS, Owens P, Sanders ME, Kuba MG, Sanchez V, Kurupi R, Moore PD, Pinto JA, Doimi FD, Gomez H, Horiuchi D, Goga A, Lehmann BD, Bauer JA, Pietenpol JA, Ross JS, Palmer GA, Yelensky R, Cronin M, Miller VA, Stephens PJ, and Arteaga CL (2014). Molecular profiling of the residual disease of triple-negative breast cancers after neoadjuvant chemotherapy identifies actionable therapeutic targets. *Cancer Discov* 4, 232-245.

Bhang HE, Ruddy DA, Krishnamurthy Radhakrishna V, Caushi JX, Zhao R, Hims MM, Singh AP, Kao I, Rakiec D, Shaw P, Balak M, Raza A, Ackley E, Keen N, Schlabach MR, Palmer M, Leary RJ, Chiang DY, Sellers WR, Michor F, Cooke VG, Korn JM, and Stegmeier F (2015). Studying clonal dynamics in response to cancer therapy using high-complexity barcoding. *Nat Med* 21, 440-448.

Echeverria GV, Ge Z, Seth S, Zhang X, Jeter-Jones S, Zhou X, Cai S, Tu Y, McCoy A, Peoples M, Sun Y, Qiu H, Chang Q, Bristow C, Carugo A, Shao J, Ma X, Harris A, Mundi P, Lau R, Ramamoorthy V, Wu Y, Alvarez MJ, Califano A, Moulder SL, Symmans WF, Marszalek JR, Heffernan TP, Chang JT, and Piwnica-Worms H (2019). Resistance to neoadjuvant chemotherapy in triple-negative breast cancer mediated by a reversible drug-tolerant state. *Sci Transl Med* 11.

Echeverria GV, Powell E, Seth S, Ge Z, Carugo A, Bristow C, Peoples M, Robinson F, Qiu H, Shao J, Jeter-Jones SL, Zhang X, Ramamoorthy V, Cai S, Wu W, Draetta G, Moulder SL, Symmans WF, Chang JT, Heffernan TP, and Piwnica-Worms H (2018). High-resolution clonal mapping of multi-organ metastasis in triple negative breast cancer. *Nat Commun* 9, 5079.

Hata AN, Niederst MJ, Archibald HL, Gomez-Caraballo M, Siddiqui FM, Mulvey HE, Maruvka YE, Ji F, Bhang HE, Krishnamurthy Radhakrishna V, Siravegna G, Hu H, Raoof S, Lockerman E, Kalsy A, Lee D, Keating CL, Ruddy DA, Damon LJ, Crystal AS, Costa C, Piotrowska Z, Bardelli A, Iafrate AJ, Sadreyev RI, Stegmeier F, Getz G, Sequist LV, Faber AC, and Engelman JA (2016). Tumor cells can follow distinct evolutionary paths to become resistant to epidermal growth factor receptor inhibition. *Nat Med* 22, 262-269.

Nguyen LV, Cox CL, Eirew P, Knapp DJ, Pellacani D, Kannan N, Carles A, Moksa M, Balani S, Shah S, Hirst M, Aparicio S, and Eaves CJ (2014). DNA barcoding reveals diverse growth kinetics of human breast tumour subclones in serially passaged xenografts. *Nat Commun* 5, 5871.

Seth S, Li CY, Ho IL, Corti D, Loponte S, Sapio L, Del Poggetto E, Yen EY, Robinson FS, Peoples M, Karpinets T, Deem AK, Kumar T, Song X, Jiang S, Kang Y, Fleming J, Kim M, Zhang J, Maitra A, Heffernan TP, Giuliani V, Genovese G, Futreal A, Draetta GF, Carugo A, and Viale A (2019). Pre-existing Functional Heterogeneity of Tumorigenic Compartment as the Origin of Chemoresistance in Pancreatic Tumors. *Cell Rep* 26, 1518-1532 e1519.

Wagenblast E, Soto M, Gutierrez-Angel S, Hartl CA, Gable AL, Maceli AR, Erard N, Williams AM, Kim SY, Dickopf S, Harrell JC, Smith AD, Perou CM, Wilkinson JE, Hannon GJ, and Knott SR (2015). A model of breast cancer heterogeneity reveals vascular mimicry as a driver of metastasis. *Nature* 520, 358-362.

Wylie AA, Schoepfer J, Jahnke W, Cowan-Jacob SW, Loo A, Furet P, Marzinzik AL, Pelle X, Donovan J, Zhu W, Buonamici S, Hassan AQ, Lombardo F, Iyer V, Palmer M, Berellini G, Dodd S, Thohan S, Bitter H, Branford S, Ross DM, Hughes TP, Petruzzelli L, Vanasse KG, Warmuth M, Hofmann F, Keen NJ, and Sellers WR (2017). The allosteric inhibitor ABL001 enables dual targeting of BCR-ABL1. *Nature* 543, 733-737.

REVIEWERS' COMMENTS:

Reviewer #1 (Remarks to the Author):

Previously, I had provided two major concerns that had not been experimentally addressed. I may accept that the points are not addressed in the manuscript. However, I feel that the decision on whether or not to accept the manuscript is then totally with the editors. The manuscript contains interesting data that should be published, however the point for me is a divergence between the claims and data.

Specifically:

- 1) Primary tumor as a model for DCCs: I agree with the arguments of the authors that the primary tumor can be a model of metastases. However, saying that does not imply that it is an adequate model for metastasis. There is simply no need to refer to metastatic disease except as a suggestion in the discussion. All I request is to down tone the link to metastasis.
- 2) Furthermore, in their conceptual answer, they even state that they CANNOT study metastasis combined with drug selection. Therefore, just say that you are studying the response to therapy and do not refer to metastasis.

The authors next address what they call a “semantic” point. It is the question on how to use the word “clone” correctly. There is a clear definition of what a clone is: “An identical copy of a DNA sequence or entire gene; one or more cells derived from and identical to a single ancestor cell OR to isolate a gene or specific sequence of DNA.” (NCI dictionary of genetic terms). My criticism was that the authors confound “identical copy of a DNA sequence” with “identical to a single ancestor” by implying that cells that differ with regard to a DNA sequence have different ancestors. By doing so, they overinterpret their data.

For example, the authors state that the question “Does each barcode mark a genetically distinct cell” would be an interesting question. As stated before, this is not an interesting question, but a very basic and fundamental question that must be answered in my opinion. Without answering this question, the author’s statements are not mirrored by experimental data. The statement in line 49-51 (each barcode reflects clonal origin, but does not imply cells with different barcodes to be different) does not rescue the point. The authors use the term “barcode complexity” but equal refer to “clonal complexity” implying cellular complexity. See example line 122-133 “This analysis revealed that primary tumors had a reduction in barcode complexity....Taken together, these results suggest that primary tumor formation is accompanied by a reduction in clonal complexity and the expansion of clones....The observation that the formation of the primary tumor is associated with a reduction in clonal complexity suggests....”).

Imagine: each barcode marks a single cell and its progeny, however, all cells are genetically identical. How would the authors interpret the barcode expansion and the enrichment of barcodes if they had this information? The current manuscript implicitly assumes that indeed each barcode marks a genetically different clone. As this is not shown in the manuscript, I insisted on either to perform experiments or to re-write the text accordingly and re-adjust the message.

This becomes also relevant for the interpretation of the MET data. The authors suggest that the finding of two different barcodes associated with MET amplification indicates de novo acquisition. However, how would they exclude that a relevant, truly clonal subpopulation (e.g. 0.1-10% of cells) harbored a MET amplification and became transduced independently. If the MET-clones are genomically different, the conclusion would be stronger.

The statement that assessment of the clonal complexity is technically not possible is not correct. As suggested by this reviewer, FISH analysis could have been done. If NGS was preferred one could have done assessment of copy number alterations (e.g. low pass sequencing of single cells of the starting population) for a glimpse into the clonal heterogeneity.

The authors state that they would have five significant findings and that only the third finding would benefit from the knowledge whether each barcode represents a genetically different cell. As outlined in my last review, I strongly disagree here. If you read the 5 points, they all depend strongly on the correct interpretation of the word "clone", because they differentiate between polyclonal recurrent and clonal recurrent tumors. If polyclonal recurrent means identical cells with different bar codes or indeed cells with different genomes makes all the difference.

Reviewer #4 (Remarks to the Author):

The authors have made useful clarifying revisions to their manuscript.

C. Eaves, PhD

Reviewer #1 (Remarks to the Author):

Previously, I had provided two major concerns that had not been experimentally addressed. I may accept that the points are not addressed in the manuscript. However, I feel that the decision on whether or not to accept the manuscript is then totally with the editors. The manuscript contains interesting data that should be published, however the point for me is a divergence between the claims and data.

Specifically:

- 1) Primary tumor as a model for DCCs: I agree with the arguments of the authors that the primary tumor can be a model of metastases. However, saying that does not imply that it is an adequate model for metastasis. There is simply no need to refer to metastatic disease except as a suggestion in the discussion. All I request is to down tone the link to metastasis.
- 2) Furthermore, in their conceptual answer, they even state that they CANNOT study metastasis combined with drug selection. Therefore, just say that you are studying the response to therapy and do not refer to metastasis.

Response: I believe that we are largely in agreement with the reviewer on this point and that our discussion about metastasis is appropriately framed.

The authors next address what they call a “semantic” point. It is the question on how to use the word “clone” correctly. There is a clear definition of what a clone is: “An identical copy of a DNA sequence or entire gene; one or more cells derived from and identical to a single ancestor cell OR to isolate a gene or specific sequence of DNA.” (NCI dictionary of genetic terms). My criticism was that the authors confound “identical copy of a DNA sequence” with “identical to a single ancestor” by implying that cells that differ with regard to a DNA sequence have different ancestors. By doing so, they overinterpret their data.

For example, the authors state that the question “Does each barcode mark a genetically distinct cell” would be an interesting question. As stated before, this is not an interesting question, but a very basic and fundamental question that must be answered in my opinion. Without answering this question, the author’s statements are not mirrored by experimental data. The statement in line 49-51 (each barcode reflects clonal origin, but does not imply cells with different barcodes to be different) does not rescue the point. The authors use the term “barcode complexity” but equal refer to “clonal complexity” implying cellular complexity. See example line 122-133 “This analysis revealed that primary tumors had a reduction in barcode complexity....Taken together, these results suggest that primary tumor formation is accompanied by a reduction in clonal complexity and the expansion of clones....The observation that the formation of the primary tumor is associated with a reduction in clonal complexity suggests....”).

Imagine: each barcode marks a single cell and its progeny, however, all cells are genetically identical. How would the authors interpret the barcode expansion and the enrichment of barcodes if they had this information? The current manuscript implicitly assumes that indeed each barcode marks a genetically different clone. As this is not shown in the manuscript, I insisted on either to perform experiments or to re-write the text accordingly and re-adjust the message.

Response:

- **Here is an example of how we interpret barcode expansion: Cells marked by barcode 8240:8518 are enriched in primary tumors, further expand in residual tumors, and completely dominate two recurrent tumors. Thus the clone marked by this barcode clearly has a stable and reproducible fitness advantage at several stages of tumor growth and recurrence. This advantage could be driven by either genetic or non-genetic (for instance, epigenetic) features of this clone. If all cells in the primary population were genetically identical, as in the reviewer’s thought experiment, then we would conclude that the stable growth advantage of this clone was driven by non-genetic mechanisms. This would not change our interpretation that the clone marked by this barcode has a stable and heritable phenotype.**

This becomes also relevant for the interpretation of the MET data. The authors suggest that the finding of two different barcodes associated with MET amplification indicates de novo acquisition. However, how would they exclude that a relevant, truly clonal subpopulation (e.g. 0.1-10% of cells) harbored a MET amplification and became transduced

independently. If the MET-clones are genomically different, the conclusion would be stronger.

Response:

- **Our results suggest that different Met-amplified clones in recurrent tumors are genetically distinct from one another, as evidenced by different Met amplicon boundaries and different copy-number alterations as revealed by WES. This argues against the model that a single Met-amplified clone in the primary tumor cell culture was transduced independently by distinct barcodes. However, proving this definitively would require isolation and genomic characterization of individual barcode-marked cells in the primary tumor population, and we maintain this is not currently technically feasible, especially since these clones are present at very small proportions in the primary cell population.**

The statement that assessment of the clonal complexity is technically not possible is not correct. As suggested by this reviewer, FISH analysis could have been done. If NGS was preferred one could have done assessment of copy number alterations (e.g. low pass sequencing of single cells of the starting population) for a glimpse into the clonal heterogeneity.

Response:

- **Our starting population has ~17000 unique barcoded clones. The approach suggested by the reviewer could work for a small number of cloned cells isolated from the starting population, but we believe is not sufficiently high-throughput to inform on the overall question of whether different clones are genetically distinct.**

The authors state that they would have five significant findings and that only the third finding would benefit from the knowledge whether each barcode represents a genetically different cell. As outlined in my last review, I strongly disagree here. If you read the 5 points, they all depend strongly on the correct interpretation of the word “clone”, because they differentiate between polyclonal recurrent and clonal recurrent tumors. If polyclonal recurrent means identical cells with different bar codes or indeed cells with different genomes makes all the difference.

Response:

- **The polyclonal recurrent tumors have an even distribution of clones: that is, no single individual clone within these polyclonal recurrent tumors is dominant. Therefore, it is not clear to me how characterizing the genome of these clones would be informative. In contrast, the interesting observation about these tumors is the observation that they recurred with thousands of evenly distributed clones (regardless of the genomic features of these subclones), while other tumors recurred with a small number of clones.**

Reviewer #4 (Remarks to the Author):

The authors have made useful clarifying revisions to their manuscript.

C. Eaves, PhD

Response:

- **We appreciate the reviewer’s remark that our revisions have improved the paper’s clarity.**